# Modularity and predicted functions of the global sponge-microbiome network

Miguel Lurgi [1], Torsten Thomas [2], Bernd Wemheuer[2], Nicole S. Webster [3,4] & Jose M. Montoya [1]

Defining the organisation of species interaction networks and unveiling the processes behind their assembly is fundamental to understanding patterns of biodiversity, community stability and ecosystem functioning. Marine sponges host complex communities of microorganisms that contribute to their health and survival, yet the mechanisms behind microbiome assembly are largely unknown. We present the global marine sponge–microbiome network and reveal a modular organisation in both community structure and function. Modules are linked by a few sponge species that share microbes with other species around the world. Further, we provide evidence that abiotic factors influence the structuring of the sponge microbiome when considering all microbes present, but biotic interactions drive the assembly of more intimately associated 'core' microorganisms. These findings suggest that both ecological and evolutionary processes are at play in host-microbe network assembly. We expect mechanisms behind microbiome assembly to be consistent across multicellular hosts throughout the tree of life.

[1] Centre for Biodiversity Theory and Modelling, Theoretical and Experimental Ecology Station, CNRS-Paul Sabatier University, 09200 Moulis, France. [2] Centre for Marine Bio-Innovation, School of Biological, Earth and Environmental Sciences, University of New South Wales, Sydney, NSW 2052, Australia. [3] Australian Institute of Marine Science, Townsville, QLD 4816, Australia. [4] Australian Centre for Ecogenomics, University of Queensland, Brisbane, QLD 4072, Australia. Correspondence and requests for materials should be addressed to M.L. (email: miguel.lurgi@sete.cnrs.fr)

Tangled webs of myriads of interacting species, as imagined and observed by Darwin[1], are ubiquitous on Earth. Their structural organisation is fundamental to understanding biodiversity patterns, community stability, and ecosystem function. Uncovering these interaction patterns and the mechanisms behind their emergence has thus been a primary focus of ecological research during the last decades[2,3]. Host-associated microbial ecosystems, ranging from invertebrates to humans, constitute excellent examples of these complex networks of ecological interactions, as they often comprise hundreds or thousands of distinct microbial types, or operational taxonomic units (OTUs)[4]. However, despite their critical role in maintaining host health, survival and function[5,6], the mechanisms behind the assembly and structuring of these microbiomes remain largely unknown.

The diversity and composition of communities is thought to be driven by the combination of two main factors: (i) which species arrive, and (ii) whether those that arrive manage to stay[7]. The ability of a species to stay in a given location is mainly determined by two factors: environment and biotic interactions[8]. In marine free-living planktonic communities, processes such as niche partitioning, environmental filtering and/or competition have a large influence on community assembly[8–10]. Recent evidence suggests that temperature and depth are the main drivers of the structure of the global ocean microbiome[11]. However, for more intimate associations such as symbioses, one would expect a strong microbial community differentiation to emerge across host species. Given the timescales involved in the development of these associations and the resulting differentiation, evolutionary dynamics are also expected to have a role in their structuring. In fact, host-associated microbiomes such as those found in the mammalian gut are structured by diverse mechanisms ranging from 'ecological' drivers, such as diet[12,13], to co-evolution[14] and co-speciation[13].

If collections of host species tend to be associated with similar microbial communities, then modules of host and microbial taxa that are found together more often than with other taxa in the community are expected to emerge. The exact composition of these modules could be mediated by both exogenous and/or endogenous factors. For instance, host-associated microbial communities may maintain an ancestral signal of host evolution, which can result in related hosts having more similar microbiomes than phylogenetically divergent ones, a pattern known as phylosymbiosis[15]. On the other hand, the way in which hosts acquire environmental microorganisms affects microbial assembly and specificity, lending support to the hypothesis that groups of species sharing physiological or behavioural characteristics can harbour similar microbial communities. For example, in humans, diet and lifestyle can modulate the abundance and metabolic activity of various members of the gut microbiota[16,17]. In marine sponges, host type (i.e. whether it is capable of harbouring highly abundant microbial communities) is a good indicator of microbiome diversity[18,19]. We should thus expect certain hosts' characteristics to have a role in the emergence of community organisation in host–microbiome networks.

Marine sponges are suitable models for exploring microbiome assembly in host-associated systems as they can harbour dense and diverse microbial communities[20] with microbial richness around $10^3$ for both high- and low-microbial abundance (HMA and LMA) species[18,21], consistent with the most microbially diverse environments on the planet[4]. Importantly, metabolic functions performed by these microbes underpin host health and survival[22]. Several lines of evidence suggest that microbial selection within sponges should be driven by factors beyond stochasticity or environmental filtering provided by the water column. Firstly, sponge–microbiome networks display several structural features known to be a hallmark of non-random

organisation across different types of complex ecological networks such as scale-free degree distributions[3,23,24]. Secondly, there is an apparent dichotomy of sponge host type between high- and low-microbial abundance species (HMA vs. LMA). In general, HMA sponges harbour more stable and diverse microbial communities than LMA sponges. These differences have been attributed to physiological features of the species, such as their water pumping rates[25]. Finally, the evolutionary history of the host has been shown to influence the diversity of symbiotic communities in marine sponges, while microbial composition is mainly driven by host identity[20]. Under a purely stochastic scenario, such as, assembly via water column filtering, neither microbiome organisation patterns nor different host strategies (i.e. HMA vs. LMA) would be likely to occur.

One of the key structural features of complex networks is their modularity[26]. The identification of modules in complex networks has revealed not only the structured organisation of these networks into tightly linked communities, but perhaps more importantly, the link between this organisation and their functioning and robustness. For instance, the modular organisation of metabolic networks has implications for information processing capabilities (i.e. sharing metabolic products across different pathways), and their robustness against the failure of specific metabolic reactions[27]. Similarly, marine food webs that result from the connection between seemingly isolated communities from the pelagic and benthic zones[28], points to the role of modularity in maintaining food web structure and robustness to perturbations such as species loss. In sponge–microbiome networks, a modular structure can reveal common metabolic pathways across host species (i.e. equivalent information processing capabilities). This can in turn facilitate the understanding of metabolic collapse in given species by looking at other species in the same module (i.e. robustness). A better understanding of the modularity of host-microbes interaction networks is thus essential to uncover fundamental patterns in their structure and function. In addition, deciphering the organisation of a global host–microbiome network of ecological interactions can facilitate analysis of connectivity profiles across host species[29], providing insight into how microbiome assembly influences microbiome structure. For instance, if HMA sponges tend to be more connected within their module relative to LMA sponges, this would suggest that the microbiome assembly processes in HMA sponges affect the structure of the network in predictable ways. On the other hand, if connectivity profiles are homogeneously distributed across hosts, this would provide evidence to support the idea that microbiome assembly processes do not affect host–microbiome network structure.

Here, we use the Sponge Microbiome Project dataset[30], comprising over 2000 microbial community samples derived from over 150 globally distributed marine sponge species, to construct the global sponge–microbiome network. We interrogate this network to analyse host occurrence patterns of ~375,000 sponge-derived microbial OTUs across these hosts. To examine the structuring of the most prevalent interactions between sponges and microbes, we also construct a sponge core–microbiome network where we restrict our analyses to OTUs that are present in at least two-thirds of the samples and with a relative abundance >0.01% across samples of the sponge species they are associated with[31]. Further, a prediction of functional profiles of the sponge core–microbiome network is conducted to gain insights into functional differentiation across network modules. We analyse the structure of these complex ecological networks and use phylogenies and environmental data to reveal whether: (i) the global network is organised into modules in which host species with highly overlapping microbial communities group together, reflecting potential commonalities in the assembly of their

microbiomes, (ii) the main drivers of this modular organisation are environmental factors, host type and/or microbial evolution within hosts, and (iii) modules correspond to unique predicted functional profiles. By relating phylogenetic and environmental information to the modular structure of the network, we reveal whether ecological or evolutionary mechanisms (or both) have been involved in the assembly of these complex networks. In addition, if network modules do not correspond to unique functional profiles, this suggests that metabolic capabilities of host-associated microbiomes do not influence network structure.

## Results

**Drivers of modularity of the global sponge–microbiome web**. To calculate the modularity of the sponge–microbiome network, we used the modularity function for bipartite networks ($Q$) proposed by Barber[32] (Methods). $Q$ is a measure of the fraction of links found between nodes within the same module to those connecting different modules. This modularity analysis finds an optimal nodes partition into modules that maximises $Q$.

Modularity analysis of the global sponge–microbiome network (hereafter global network) detected 5 modules. This modular partition is robust to data rarefaction performed independently 100 times to the number of reads in the smallest sample ($10^4$ reads). Subsequent analyses are, therefore, not affected by biases in sequencing depth (Supplementary Table 6). Modules were heterogeneous in terms of diversity, with module 1 comprising 8 sponge species and 5276 OTUs, module 2 including 9 sponges and 43,971 OTUs, module 3 comprising 46 sponges and 86,803 OTUs and modules 4 and 5 including 14 and 79 sponge species and 53,875 and 184,943 OTUs, respectively.

Our results suggest that the modular structure of the global network is likely determined by two ecological factors: (i) environment (via water temperature), and, to a lesser extent, (ii) host type, i.e. LMA vs. HMA (Fig. 1 and Supplementary Table 1). Geographical location (ecoregion) and depth had no influence on the modular organisation of the global network (Fig. 1 and Supplementary Table 2). The modules that emerged can be grouped into warm (27.16 ± 2.82 °C, for modules 2 and 3) and cold (18.77 ± 5.15 °C for modules 4 and 5) water modules. Within each temperature range further categorisation was evident based on microbial abundance within the host (host type), except for the warm water LMA group, for which half the sponges were HMA (Fig. 1).

A set of sponges laid outside of this classification (module 1), the majority coming from a single sampling location (Breaker Bay, Wellington, New Zealand). Sponges within module 1 had a high proportion of OTUs belonging to the phylum Firmicutes (22% of OTUs in module 1 compared to an average 1% across the remaining modules, see Supplementary Figure 1). Although the identification of this module substantiated the robustness of the modularity algorithm, further investigation was undertaken as Firmicutes are generally at low abundance in healthy sponge microbiomes and are instead more often associated with diseased/stressed or contaminated individuals[33–36]. These further enquiries indicated that sponges from this location have probably been contaminated from external sources (personal communication from Dr Mike Taylor, University of Auckland, New Zealand).

Interesting differences were observed in the taxonomic composition of OTUs across modules of the global network (Supplementary Figure 1). For example, the microbiomes of modules 3 and 4 (those comprised mainly of HMA sponges) contain a higher fraction of unclassified Bacteria than other modules, while module 5 harboured a larger fraction of Alphaproteobacteria (Supplementary Figure 1). This suggests that not only ecologically but also genetically related microbes are

forming microbial associations within modules. In addition to ecological factors, evolutionary processes affect the modular structure of the global network. A microbial phylogenetic signal was evident in the modular organisation of OTUs in the global network (UniFrac score = 0.757989, $p$-value < 0.001 and see Supplementary Table 3 for full pairwise comparisons), supporting the notion that specific host-associated microbial communities have evolved within (or in association with) groups of sponge species that harbour phylogenetically similar microbiomes.

**Modules of the sponge core–microbiome match host type**. We further restricted the analysis to core microbial OTUs that corresponded to those found in at least two-thirds of the samples and with a relative abundance >0.01% across samples of the sponge species they were associated with. Despite the application of such strict constraints on the definition of the core microbiome (Methods), a fully connected sponge core–microbiome network (hereafter core network) remained (i.e. all sponge species and OTUs in the network remained part of a single coherent connected component). Within this much smaller (only 10,941 OTUs and 51,207 links) network, we found seven modules. The modular structure of the core network was less influenced by environmental factors than that of the global network (Supplementary Figure 2). The majority of HMA sponges (73.53% or 25 out of 34), grouped together in the same module (number 4), whereas LMA sponges were distributed across several modules based on similarity in their microbial composition (Fig. 2, Supplementary Figure 2 and Supplementary Table 4). The distinction between HMA and LMA species seems clear given that almost all HMA species grouped together. This is also evident when looking at the taxonomic profile of microbiomes across modules (Supplementary Figure 3), where, as was the case of the global network, the module containing the majority of HMA sponges showed a microbiome dominated by unclassified Bacteria. In contrast, the splitting of LMA species across several modules indicates that, even though there is a distinct difference between HMA and LMA sponges based on microbial abundance, there seems to be a larger number of distinct OTU combinations or communities that can interact/are found within LMA sponges. This is particularly evident in genera such as *Haliclona*, whose members are found across many modules of the core network, and which also show high variability in the fraction of their OTUs that are exclusive to the network module they belong to (Fig. 3). This suggests that this genus of LMA sponges possesses a highly diverse microbiome that is distinct across its constituent species, while at the same time being shared with many other sponges from different phylogenetic backgrounds.

The majority of HMA sponge species (74.51%) had at least 70% of their links within their corresponding module, with a participation coefficient $P < 0.5$, where $P$ is a weighted measure of the extent to which sponges harbour OTUs mainly found in other modules. In contrast, LMA species showed greater variability in the fraction of links contained within their module, with a large fraction of these sponges (83.12%) having high participation coefficients ($P > 0.4$) (Fig. 4). This suggests that HMA sponges are more similar in terms of microbial community composition than LMA sponges, which can harbour microbes that are shared across many different sponge species. The qualitative result from the network analysis was quantitatively confirmed with microbial community composition data. Multivariate analysis of variance performed on the microbial community distance matrix between hosts revealed significant differences across host types ($F = 19.78$, $p$-value = 0.001). Further, analyses of the dispersion of variance within groups revealed that microbial communities within HMA sponges are more similar among themselves than those found in

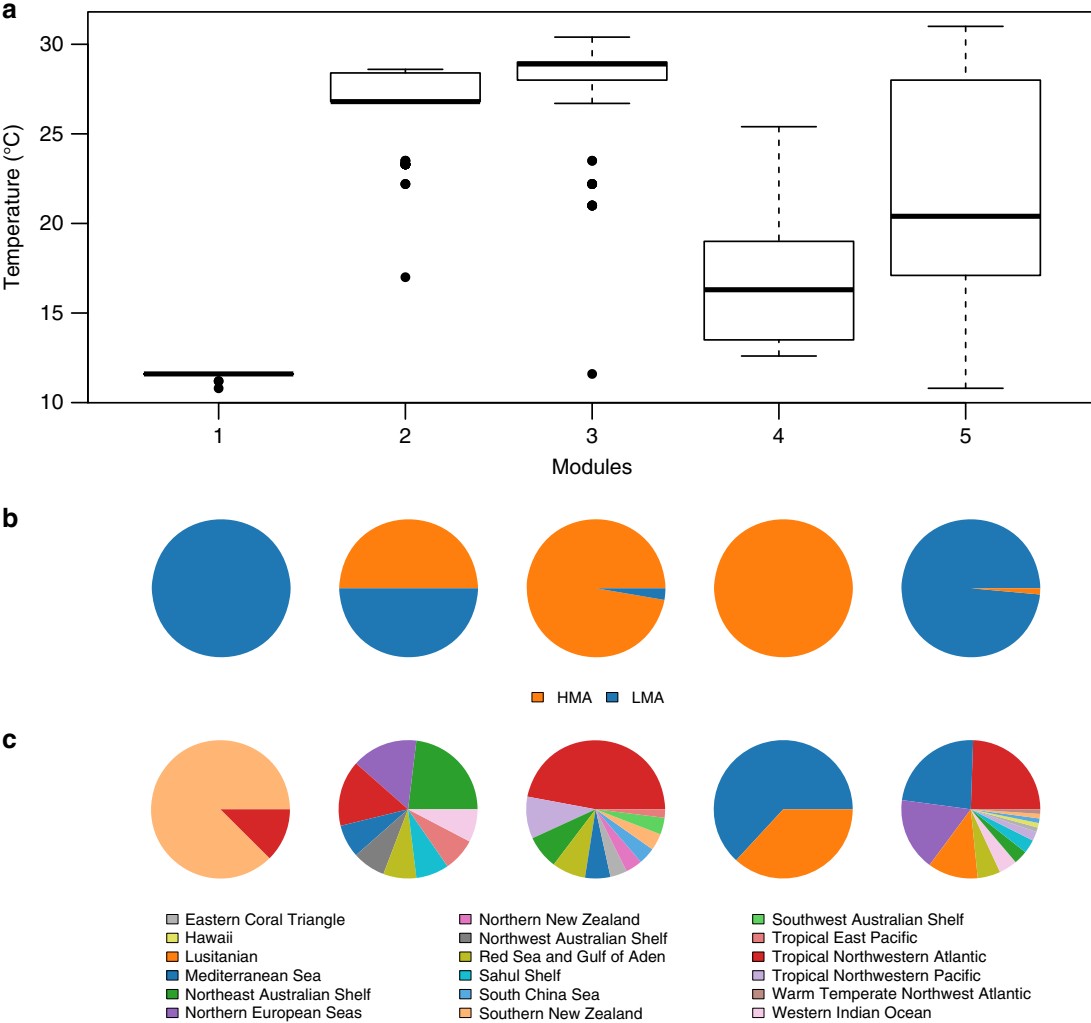

**Fig. 1** Drivers of modularity of the global sponge–microbiome network. **a** Distribution of temperature (in °C) from sponge sampling sites across network modules identified at the whole network level. Thick line in the middle of boxes represents the median values; box limits are lower and upper quartiles; whiskers, 1.5× interquartile range; points, outliers. **b** Composition of host type (high- vs. low-microbial abundance sponges, HMA and LMA respectively) per module. **c** Composition of marine ecoregions from where the samples were collected within each module. Labels for the modules on the x axis in **a** apply to **b** and **c**. Data use to generate this figure are available as Supplementary Data 1

LMA sponges (average distance to centroid = 0.47 for HMA vs. 0.64 for LMA).

**Microbial phylogeny and the core–microbiome network**. In addition to sponge type (i.e. HMA or LMA), the assembly of the core network appears to be highly influenced by the evolution of the microbiome within their host, with microbiome phylogeny found to be a strong predictor of the core network modularity (UniFrac score = 0.816360, p-value < 0.001 and see Table 1 for full pairwise comparisons). Evolution of the microbiome within its host is also supported by evidence of high specificity of microbial composition across host species (Supplementary Figure 4). This suggests that modules share microbial assemblages that correspond to similar evolutionary trajectories, which is reflected in the microbial phylogenies.

Surprisingly, the phylogenetic background of sponge hosts does not have a role in the organisation of the core network. We found no relationship between sponge phylogeny and network modules (p-value > 0.1 for all UniFrac pairwise module comparisons; Mantel r = 0.1186), suggesting a weak influence of host phylogenetic relatedness on microbiome assembly. Nonetheless,

the majority of HMA sponges that grouped into the 'HMA module' were closely related within their phylogenetic tree (Fig. 2). This trend was also observed, although to a lesser extent, in other modules (e.g. 6 or 7) (Fig. 2).

Importantly, when looking at microbial OTUs taking the role of provincial hubs (i.e. those with a high within-module degree ($z > 2.5$) and a small participation coefficient ($P < 0.3$)), we found that the phylogenetic signal of the modular partition for this subset of microbes was weaker than that observed when all microbes were considered (see Table 1 for UniFrac results). This suggests that, even though evolutionary processes play an important role in structuring microbial communities within modules, sponges are able to acquire and maintain microbes that are not necessarily phylogenetically related to other members of their associated microbial community. To assess whether the module membership of provincial hub OTUs can be linked to ecoregion, and hence environmental specificity, we investigated their geographical origin. Most of these provincial hubs (72.73%) are found in more than 5 bioregions within their module, while only 14.29% are exclusive to a single bioregion. This suggests that these 'connecting' OTUs might have been acquired and conserved by different sponge lineages regardless of their location.

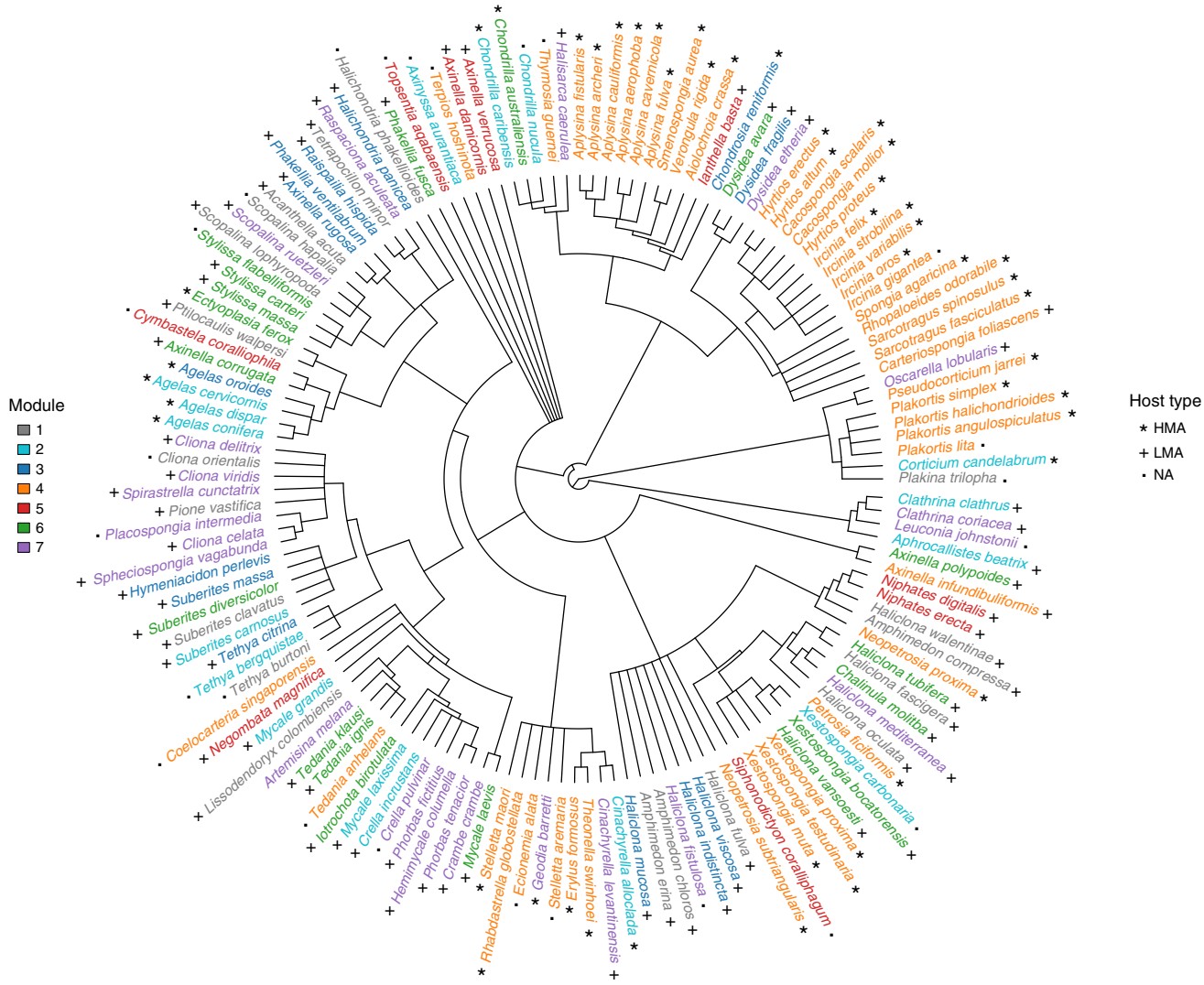

**Fig. 2** Sponge phylogeny is associated with core network modularity. Sponge species belonging to the same module identified over the sponge core–microbiome network display the same colour. Stars, crosses and filled squares refer to sponge type according to whether they are high- or low-microbial abundance species (HMA and LMA respectively). NA = no information for host type is available. Only 144 out of the 156 host species considered in this study are shown because 10 of the original host species were not resolved to the species level and the sponge species *Ircinia fasciculata* and *Strongylacidon conulosa* did not have matches in the Open Tree of Life from which this tree was extracted (see Methods). For a complete list of sponges, their module membership and types see Supplementary Table 4

**Functional profiling of microbial modules**. Prediction of the functional profile of the 'core microbiome' revealed a splitting based on functionality provided by the microbial communities within each module (adjusted p-values < 0.05 for all PERMANOVA pairwise comparisons involving modules 3, 4, 7, see Table 2 for all results). HMA-dominated module 4 was predicted to be the most functionally distinct (F = 24.09 ± 11.31 for all pairwise comparisons involving this module, compared to the overall F = 12.26 ± 11.31), suggesting a marked functional differentiation between HMA and LMA species. Nonetheless, differentiation across LMA-dominated modules was also observed, with some predicted to be more functionally similar to module 4 than others (e.g. F = 16.04 for 4 vs. 1, and F = 44.52 for 4 vs. 6). As expected, module 2, comprising only 50% HMA sponges, seems to be more functionally similar to other modules than the HMA-dominated module 4 (Table 2). No clear distinction in predicted functional profiles was found amongst modules 1, 2, 5

and 6 (Table 2). Their members were thus grouped into a single super-module in subsequent functional analyses.

Indicator analysis confirmed functional differentiation between modules in the core network (adjusted p-value from permutation tests < 0.05 for 86 of the 435 pathways identified for pairwise comparisons of metabolic pathways across modules); even if a certain degree of functional overlap was observed between modules (Supplementary Figure 5). 12 (or 14%) of the functions that differentiated between modules included pathways for the production of bioactive secondary metabolites; for example, streptomycin biosynthesis, polyketide sugar unit biosynthesis, sesquiterpenoid and triterpenoid biosynthesis, isoflavonoid biosynthesis, antibiotic biosynthesis, prodigiosin biosynthesis and biosynthesis of enediyne antibiotics (Supplementary Figure 5). Sponges and their microbial symbionts are renowned for their bioactive compound production[37,38]; and this function appears to have a significant role in structuring the microbiome.

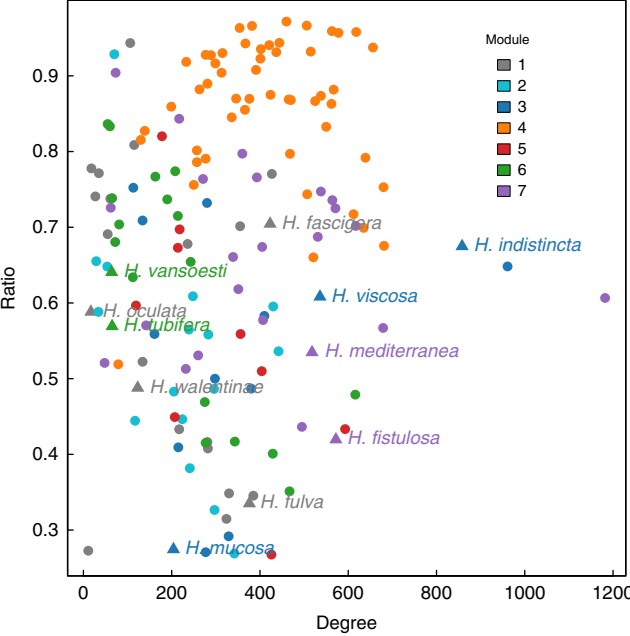

**Fig. 3** Degree–ratio relationships for sponges in the core network. Degree (i.e. number of OTUs found within the host) vs. ratio of within-module links vs. across-module links of the sponge species in the sponge core–microbiome network, based on the core microbiome modularity. Each point corresponds to a sponge species. Different colours represent species memberships in the modular classification. *Haliclona* species are highlighted (with species name and a triangle) to illustrate the degree of heterogeneity, both in terms of OTUs and the degree of their intra-module connectivity, among low-microbial abundance species from the same genus

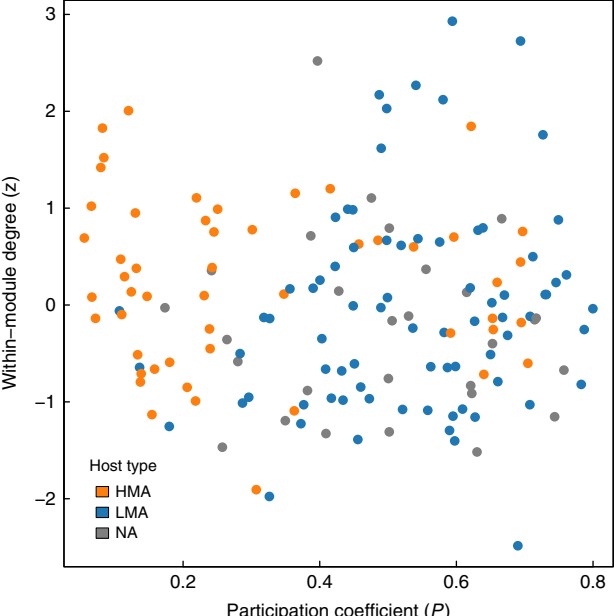

**Fig. 4** Microbial abundance drives sponges' network connectivity. Within-module degree ($z$) vs. participation coefficient ($P$) for the sponges the core network. Different colour classes show different sponge types, divided into high- and low-microbial abundant species (HMA and LMA, respectively). NA = no microbial abundance classification is available. LMA sponges generally share a greater number of OTUs with species in other modules than HMA ones, i.e. larger $P$ (KS test: $D = 0.57601$, $p$-value < 0.001). However, the extent to which sponges tend to harbour a large fraction of OTUs from those found in their module is variable across both host types

Constrained distance-based redundancy analysis showed a clear separation amongst sponges belonging to the different network modules based on the predicted functional profiles of their microbiomes (Fig. 5). Functional differentiation is evident not only between the HMA-dominated module 4 and the other modules, but also across modules comprised predominantly of LMA species (Fig. 5 and Supplementary Figure 5). Together with the strong microbial phylogenetic signal detected for modularity, this suggests that different sets of LMA sponges (and to a lesser extent HMA species given their presence in other modules) have different functional potential by harbouring specific and distinctly structured bacterial communities. Although microbiome differentiation is a better systematic predictor of the differences amongst modules than metabolic potential (Supplementary Table 5), in many cases (particularly those involving comparisons with modules 4 and 7), metabolic differentiation is stronger than what would be expected by solely looking at differences in microbiome composition (compare the *F* statistics of Table 2 with those on Supplementary Table 5).

## Discussion

The modular organisation of the global sponge–microbiome network is driven by two main factors: environment (via temperature) and, to a lesser extent, host type (via microbial abundance). When looking at the core microbiome, however, the phylogenetic origin of the microbial associates is the main driver. This microbial modularity translates into distinct predicted functional capabilities among different modules of the core microbiome. In addition, host type has a strong influence on the connectivity of sponges in the network, with HMA sponges generally sharing a large fraction of microbes with other sponges within their module. LMA sponges on the other hand, show less

overlapping microbial communities, hosting many microbes that are shared with sponge species in other modules. This finding suggests that LMA sponges can harbour a larger number of distinct microbial communities. We hypothesise that the greater differentiation observed across LMA sponges compared to HMAs is due to physiological traits related to the sponges' ability to filter seawater, such as water pumping rates[18,25], however, insufficient physiological data are currently available to test hypotheses of this kind. This therefore constitutes an important area for future research, which will shed light on the current HMA–LMA dichotomy for which we lack convincing and unequivocal mechanisms. Our results show, nonetheless, that abiotic factors can have a strong influence on the broad structuring of host-associated microbial communities, but a common microbial evolutionary origin is most likely responsible for the assembly of the core, likely symbiotic, microbiome.

Marine sponges and their microbial communities are organised into tightly linked modules. These modules are linked by a handful of sponge species that share microbes with other species around the world. The number and composition of these modules are, however, dependent on the scale of ecological organisation being considered. When all sponge-associated microbes are included, modular organisation is broader (i.e. the network is organised into a small number of modules) and associated to abiotic factors, primarily seawater temperature. Importantly, temperature has also been suggested to be a major driving factor in the assembly of planktonic communities[11], suggesting that the global network could be heavily influenced by transient microbes. This seems to be the case, since when the sponge core–microbiome network is considered (i.e. when transient microbes are removed), the relationship between temperature and modularity is lost. In fact, modules in the core–microbiome

**Table 1 Microbial phylogenetic signal in the modularity of the sponge core–microbiome network**

| Pairs | UWScore | p-value (UWSig) | UWScore—prov | p-value—prov |
|---|---|---|---|---|
| 1-2 | **0.903157** | <0.001 | | |
| 1-3 | **0.878039** | <0.001 | | |
| 2-3 | **0.890744** | <0.001 | 0.96886 | 0.327 |
| 1-4 | **0.933797** | <0.001 | | |
| 2-4 | **0.888006** | <0.001 | **0.932771** | 0.01801 |
| 3-4 | **0.914062** | <0.001 | **0.937423** | 0.003009 |
| 1-5 | **0.89443** | <0.001 | | |
| 2-5 | **0.89295** | <0.001 | | |
| 3-5 | **0.901336** | <0.001 | | |
| 4-5 | **0.916223** | <0.001 | | |
| 1-6 | **0.889275** | <0.001 | | |
| 2-6 | **0.882286** | <0.001 | 0.981931 | 0.162 |
| 3-6 | **0.889612** | <0.001 | 0.939094 | 0.07801 |
| 4-6 | **0.904889** | <0.001 | **0.966007** | < 0.001 |
| 5-6 | **0.880169** | <0.001 | | |
| 1-7 | **0.892254** | <0.001 | | |
| 2-7 | **0.891977** | <0.001 | 0.881778 | 0.534 |
| 3-7 | **0.867124** | <0.001 | 0.971572 | 0.324 |
| 4-7 | **0.905576** | <0.001 | **0.964441** | 0.005009 |
| 5-7 | **0.909807** | <0.001 | | |
| 6-7 | **0.891617** | <0.001 | 0.979531 | 0.124 |

Results from unweighted UniFrac analyses used to assess the significance of the phylogenetic fingerprint of the sponge-associated microbial communities in the modular structure detected in the sponge core–microbiome network. Second and third columns show results for all OTUs found in the core microbiome, whereas the third and fourth columns (identified by 'prov') show results for the subset of 'provincial hub' microbes, i.e. those with a high within-module degree ($z$) and a small participation coefficient ($P$). See Methods for an explanation of these terms. Empty cells represent situations where modules identified in the network had no provincial hubs. All significant scores (UniFrac $p$-value < 0.05) are highlighted in bold

network are constituted by microbes that are more phylogenetically similar than would be expected by chance. This suggests that an evolutionary assembly of microbial communities, emerging from closely related microbial phylogenetic lineages across groups of sponge species has left a fingerprint on the structure of the core network.

In addition to environmental factors, network modules tend to match host type. One of the core network modules is comprised almost entirely of HMA species, while the LMA species distribute across the remaining modules. On the other hand, microbial provincial hubs (i.e. those OTUs with a large degree of within, but small between module connectivity) do not show a phylogenetic signal in their modular arrangement. This suggests that sponges can acquire and maintain highly generalist microbes that are not necessarily phylogenetically related to the rest of the microbes in their module. The potential acquisition of these members of the microbiome highlights the likely importance of ecological processes such as environmental/horizontal transmission of microbes for the structuring of the sponge 'core microbiome'. Provincial hubs act as connectors within the network, enhancing the proximity of nodes within each module to each other. In sponge microbiomes, this 'closeness' is likely related to shared metabolic functions, suggesting that provincial hub microbes act as facilitators of common functions across hosts within modules.

Even though our work represents an important first step towards an improved understanding of causes and consequences of the assembly of the sponge microbiome, deeper insights could be gained if optimised methods for analysing/processing data were developed. For example, in processing the data, we filtered OTUs present in the microbiome by removing those that were common in the surrounding seawater. Whilst there was strong justification for this approach, it also represents a potential caveat, as genuine members of the sponge microbiome may get removed in the process.

Our results are consistent with the classical hypothesis of a widespread distribution of microbes across the globe, from which particular environments (or in this case sponges) select specific assemblages[8], a notion which has also recently found empirical support in free-living planktonic communities[11]. Sponges, due to their ability to filter large volumes of seawater, are exposed to high levels of microbial influx from the environment[39]. Some of these microbes may be enriched and grow to higher abundances inside the sponge host and become established as permanent populations, forming microbial communities that appear specific to their hosts[20]. In addition, some sponge-associated microbes which are difficult to detect in the surrounding seawater can chemotax towards sponge-derived compounds, indicating an active role of the microorganisms in initiating the species-specific partnerships[40]. Even though at the phylum level there might considerable taxonomic overlap between free-living and sponge-associated microbial communities, at the OTU or genus level, we can expect limited overlap between them[4].

Starting from a given set of initial species, free-living microbial communities might assemble in different ways. The outcome of this assembly process can depend on several factors, ranging from environmental conditions and biotic interactions[41], the spatial arrangement of individual microbes[42] to meta-community processes[43]. These factors together provide a multi-stability landscape in which different basins of attraction (i.e. alternative community assembly trajectories) might yield different microbial assemblages, even from the same original set of species[43,44]. By contrast, microbes living inside multi-cellular hosts are more likely to be constrained by the environment in which they live and the survival needs of the host[45]. In these cases, (co-) evolutionary processes become pivotal. In marine sponges in particular, although not exclusively, co-evolution is reinforced by the vertical and horizontal transmission of at least a subset of microbes from parents to offspring[46–49]. In the case of vertical transmission, an evolutionary priority effect can emerge, by which an evolving community of microbes is inoculated from a selected set of available taxa. Our results suggest that this is true not only at the local scale (i.e. within-sponge species), as has been previously found[24], but also for the global core network. Even though our results do not conclusively show an influence of co-evolutionary processes in the assembly of the core network, given the lack of relationship between sponge phylogeny and network modular structure, co-evolution at the host level could still be occurring. Further research to unveil phylogenetic fingerprints left behind by co-evolution and co-diversification processes should focus on subsets of microbes displaying obligate or quasi-obligate relationships with their hosts, e.g. vertically transmitted symbionts.

Modules of the sponge core–microbiome network had distinct predicted functionality, with microbial communities within at least three of the core network modules having significantly different metabolic functions. This suggests that different groups of host species have selected distinct microbiomes that have the metabolic functions required for their survival. Interestingly, many of the functions that differed across modules in the core network were associated with the production of bioactive compounds. Bioactive secondary metabolites are commonly produced by sponges and/or their associated microbes[37,38] and are thought to have a major role in mediating host–microbiome interactions and/or in chemical defence[45]. Thus, if hosts facing analogous biological challenges maintain similar microbiomes to help them meet these challenges, a modular structure in the host–microbiome network can emerge. This organisation can in turn feedback as a selective pressure into the evolution of the

**Table 2 Heterogeneity in functional profiling of modules in the sponge core–microbiome network**

| Pairs | F statistic | R² | p-value | Adjusted p-value | Significance |
|---|---|---|---|---|---|
| 1 vs. 2 | 1.372646 | 0.03672862 | 0.188 | 0.685327436 | |
| 1 vs. 3 | 3.345947 | 0.09205832 | 0.009 | 0.045931520 | . |
| 1 vs. 4 | 16.036242 | 0.18215501 | 0.001 | 0.006379378 | * |
| 1 vs. 7 | 9.448082 | 0.18014161 | 0.001 | 0.006379378 | * |
| 1 vs. 6 | 1.822294 | 0.04463967 | 0.080 | 0.340233479 | |
| 1 vs. 5 | 1.403918 | 0.04617557 | 0.186 | 0.685327436 | |
| 2 vs. 3 | 5.941705 | 0.18037029 | 0.003 | 0.016404114 | . |
| 2 vs. 4 | 23.547491 | 0.26296093 | 0.001 | 0.006379378 | * |
| 2 vs. 7 | 16.040983 | 0.30242620 | 0.001 | 0.006379378 | * |
| 2 vs. 6 | 3.107370 | 0.08605917 | 0.003 | 0.016404114 | . |
| 2 vs. 5 | 1.525886 | 0.06221531 | 0.138 | 0.556013133 | |
| 3 vs. 4 | 13.359251 | 0.17495262 | 0.001 | 0.006379378 | * |
| 3 vs. 7 | 3.912138 | 0.10318959 | 0.010 | 0.047845333 | . |
| 3 vs. 6 | 10.089029 | 0.25166559 | 0.001 | 0.006379378 | * |
| 3 vs. 5 | 7.565526 | 0.27445606 | 0.001 | 0.006379378 | * |
| 4 vs. 7 | 18.998684 | 0.20651039 | 0.001 | 0.006379378 | * |
| 4 vs. 6 | 44.524645 | 0.39220246 | 0.001 | 0.006379378 | * |
| 4 vs. 5 | 28.084935 | 0.32250050 | 0.001 | 0.006379378 | * |
| 7 vs. 6 | 24.190476 | 0.37685460 | 0.001 | 0.006379378 | * |
| 7 vs. 5 | 20.280478 | 0.40334696 | 0.001 | 0.006379378 | * |
| 6 vs. 5 | 2.902114 | 0.10041183 | 0.022 | 0.099067984 | |

Results of pairwise permutational analysis of variance (PERMANOVA) performed over the set of metabolic functions inferred for each module in the sponge core–microbiome network based on the microbiome. Significance codes: '*' 0.01, '.' 0.05, ' ' 0.1

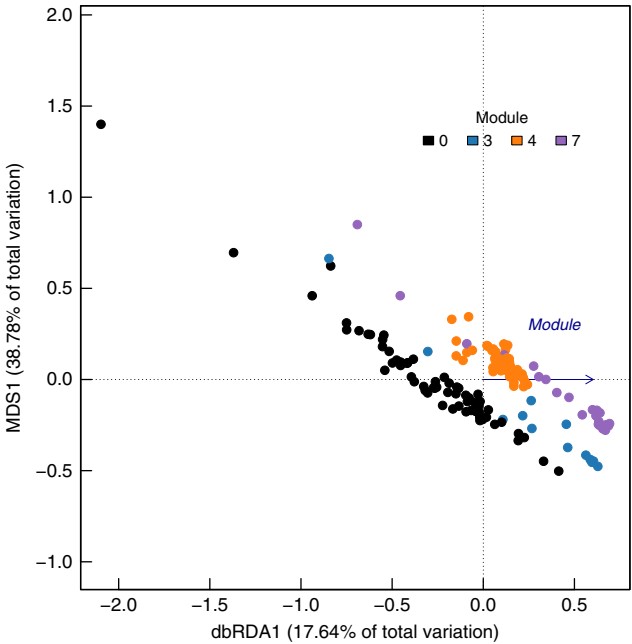

**Fig. 5** Modules of the core network are functionally distinct. Constrained distance-based redundancy analysis shows a clear separation of the functional profiles of sponges belonging to different network modules. Module was the sole constraining variable for this analysis and hence the second dimension comes from unconstrained ordination. The axis derived from the modular classification of the sponges (dbRDA1) explains 17.64% of the total variation, while the first axis of unconstrained ordination explains 38.78% of the variation. Module 0 (black dots) comprises original modules 1, 2, 5 and 6 in the sponge core–microbiome network, which were not found to be significantly different from each other by pairwise PERMANOVA comparison in terms of predicted metabolic functions

microbiome, completing an eco-evolutionary loop that influences the assembly of host–microbiome networks[50,51]. HMA sponges exemplify this. They cluster together into a single module of the core network, which suggests that eco-evolutionary dynamics lead to functional equivalence, which has been linked to evolutionary convergence in the sponge microbiome[52]. Importantly, we found that the microbiomes of HMA-dominated modules harbour a high fraction of unclassified bacteria. Further research is needed to identify and classify these bacterial strains to fully understand their metabolic potential.

We found that the phylogenetic relatedness of OTUs was linked to the modularity of both the global and core sponge–microbiome networks. A similar pattern has been reported for a smaller number of sponge species harbouring phylogenetically distinct microbial communities[53], although only at the individual sponge level. Interestingly, Ghoul and Mitri[54] argue that phylogenetic relatedness in microbial communities is expected to be selected against due to similarities in metabolic needs; i.e. phylogenetically related microbes would be more likely to compete than non-related ones due to niche similarities. However, several ecological conditions might be involved in the manifestation of selection for competition[54]. Some of these conditions, such as high microbial diversity and invasion/migration processes, which lead to low metabolic overlap, are characteristic features of many marine sponges[55]. Similarly, the high environmental complexity of sponges can result in a highly structured microbial community through the occupation of different physical locations in the host and through the dynamic chemical environment mediated by intermittent sponge pumping[22,56]. These mechanisms can contribute to low selection for competition in microbial communities[54] and would result in a small fraction of competitive interactions amongst microbes within the host. Another possible mechanism for selection against phylogenetic relatedness in microbial communities is metabolic cross-feeding[57], where competitive pressures are alleviated because phylogenetically distant microbial taxa coexist based on their

metabolic abilities. This is consistent with recent findings by Björk et al.[58] of prevalent amensal and commensal interactions between bacteria within the core sponge microbiome of three HMA sponges. Positive interactions however, are known to be destabilising in ecological communities, including microbiomes[59]. To counteract this destabilising effect, Coyte et al.[59] proposed that competitive interactions are key for the maintenance of a stable microbiome. Hence, even though a low selection for competitive interactions should be expected in microbial communities, at least some competition is expected to occur in order to maintain microbiome stability. Thus, a mixture of interaction types, along with spatial structuring[42], seems to be necessary for the assembly and stability of host-associated microbial communities. To shed light on the configuration and nature of these microbial interactions within the sponge microbiome, future research should focus on the inference of abundance based OTU–OTU interaction networks, additionally considering environmental factors that might modulate bacterial interactions.

In conclusion, distinct organisational patterns at different ecological scales have been reported for a number of microbial systems. For instance, the network of ecological interactions in bacteria–phage communities of the Atlantic Ocean is modular at large, but nested at small spatial scales[60], pointing to different mechanisms operating at different spatial scales[61]. Sponge-associated microbial communities also display scale variability, with different modular patterns emerging when considering the entire vs. the core microbiome networks. This differentiation reflects the relative strength of ecological vs. evolutionary factors on the assembly of host–microbiome networks. The emergent pattern for sponges shows that seawater temperature constitutes a major driver in the assembly of the large-scale host-associated microbiome whereas microbial evolutionary origin plays a more prominent role in structuring the core sponge microbiome.

## Methods

**Network construction**. Using the 16S rRNA gene sequencing data from the Sponge Microbiome Project (SMP) dataset[30] which comprises marine sponge samples from many locations across the globe (Supplementary Figure 6), we selected samples belonging to sponges that were compliant with the following criteria: (i) at least three samples where present in the dataset, (ii) these had >10^4 16S rRNA gene sequencing reads, (iii) were extracted from healthy, adult individuals, and (iv) had unequivocal taxonomic identification. In addition, OTUs that were present in seawater samples from the SMP dataset with relative abundances larger than 0.01%, were removed from sponge samples, since they were likely derived from seawater that was part of the sponge sample.

From this resulting dataset, a bipartite network was constructed based on OTU occurrence (presence/absence) within each sponge species. This network was constructed by defining two sets of nodes (OTUs and sponge species) and assigning links between an OTU node and a sponge node whenever that OTU was found in a sample of the sponge species. We refer to the network thus constructed as the global sponge–microbiome network (the global network). This process yielded an ecological network composed of 156 host sponge species and 374,868 OTUs with 1,882,922 interactions between these sets of nodes. Overall network patterns such as degree distributions or degree vs. abundance relationships were obtained and matched previous analyses performed over a subset of these data (compare Supplementary Figures 7 & 8 with Figs. 6, 7 in ref. [20], respectively).

**Extracting community structure**. To extract community structure (i.e. the identification of modules) in this bipartite network, we employed the LPAwb + community detection algorithm[62]. This algorithm is based on the maximisation of the modularity function ($Q$) for bipartite networks (Eq. 1), as proposed by Barber[32].

$$Q = \frac{1}{2m} \sum_{i,j} \left( A_{ij} - P_{ij} \right) \delta(g_i, g_j) \qquad (1)$$

where $m$ is the number of edges in the network, $A_{ij}$ are the adjacency matrix elements (1 if a link between vertices $i$ and $j$ exists and 0 otherwise), $P_{ij}$ are the probabilities from a null model that an edge exists between vertices $i$ and $j$, $g_i$ is the module to which vertex $i$ belongs. $\delta(r, s) = 1$, if $r = s$ and 0 otherwise[32].

Modularity algorithms are able to obtain partitions of nodes in the network for which links between nodes inside modules are more common than across them. Considering the computational resource utilisation of the algorithm employed to

perform this task, we ran the modularity analysis by discarding OTUs that had only one link (i.e. they were present in only one sponge). While this might affect the absolute value of the modularity measure ($Q$), it does not affect the way in which nodes are partitioned into modules, since all nodes that are connected to only one other would always be assigned to the same module its partner belongs to. From a biological point of view, an OTU found in only one sponge species (i.e. nodes with one link) can only be part of the module to which its only host belongs. We thus consider the modular partition of the network obtained in this way to be equivalent to that of the whole network.

**Data rarefaction**. To assess the robustness of the modularity partition obtained using the procedure described above to a different sequencing depth, we performed pairwise permutational analyses of variance (PERMANOVA) on 100 rarefied datasets of the original OTU table using the same modules detected in our network as an independent variable. Thus, testing for 'network module' as a source of variation in microbiome composition. In this way, we can assess whether the modular partition obtained for the network (i.e. the main topological feature our analyses focus on) is robust across the rarefied datasets.

We rarefied the data to 10^4 reads per sample (i.e. the size of the smallest sample in the dataset) using the rrarefy function of the vegan[63] package. This procedure was repeated independently 100 times to obtain 100 different rarefied instances (i.e. randomisations) of the data. We then analysed each of these rarefied datasets using pairwise PERMANOVA with the adonis function from vegan[63]. p-values were corrected for false discovery rates due to multiple comparisons using the Benjamini–Yekutieli correction. We then obtained the average and standard deviation of the values obtained from this test across the 100 randomisations of the rarefied data.

**Defining species roles in the network**. Using information obtained from the modularity algorithm, we further evaluated the structure of the communities by looking at the position of the different nodes in the network. We employed a set of measures based on the links of the nodes and their connectivity to other modules to determine node connectivity profiles: the nodes' degree (i.e. number of links), the ratio of intra-module vs. extra-module links, and the fraction of nodes they are connected to in their module.

This connectivity profile was complemented using two, more general measures that are normalised by the number of nodes and links in other modules: the within-module degree ($z$, Eq. 2) and the participation coefficient ($P$, Eq. 3), as proposed by Guimerà and Amaral[29].

$$z_i = \frac{k_i - \overline{k_{Sl}}}{\sigma_{k_{Si}}} \qquad (2)$$

where $k_i$ is the number of links of node $i$ has to other nodes in its module $s_i$, $\overline{k_{Sl}}$ is the average of $k$ over all nodes in $s_i$, and $\sigma_{kSi}$ is the standard deviation of $k$ in $s_i$.

$$P_i = 1 - \sum_{s=1}^{N_M} \left( \frac{k_{is}}{k_i} \right)^2 \qquad (3)$$

where $k_{is}$ is the number of links node $i$ has to nodes in module $s$, and $k_i$ is the total degree of node $i$. These measures, as their names suggest, provide a quantified measure of the degree to which a node is well connected within its module and the extent to which it connects to other modules in the network[29].

**Impact of environmental variables and microbial abundance**. The effect of the environment on the structure of the global network was assessed by looking at the relationships between (i) temperature, (ii) depth, and (iii) biogeographical location to the structural modularity of the network, using environmental metadata derived from the SMP dataset. Biogeography was assessed by mapping the sampling location of each sponge sample onto the Marine Ecoregions of the World[64]. These marine ecoregions represent broad-scale patterns of species and communities in the ocean such that communities within ecoregions are thought to have been subjected to similar historical conditions (environmental or otherwise) that could have influenced their assembly[64]. Biogeographic data was obtained as a Geographical Information System (GIS) layer from the World Wildlife Fund (https://www.worldwildlife.org/publications/marine-ecoregions-of-the-world-a-bioregionalization-of-coastal-and-shelf-areas) and overlaid on sampling points for each sponge species available as part of the SMP dataset. Finally, information on microbial abundance within each host species was used to classify ~83% ($n = 129$) of the species into either a high-microbial abundance (HMA) or low-microbial abundance (LMA) category as per[18,65] (see Supplementary Data 1 for the metadata used in this study).

**Defining the core microbiome**. Sponge-associated microbial communities comprise different types of microorganisms according to the intimacy and repeatability of the association with their host, forming a continuum from core through transient to opportunistic microbes[55,58]. The recognition of this distinction among bacterial taxa within hosts has prompted the development of the 'core microbiome' concept as a way of identifying microbes that are consistently found within a

particular sponge species, likely perform core metabolic functions and therefore eventually become symbionts[22,31,52]. To reveal interaction patterns between marine sponges and the microbes that are in close association with them, we also constructed and analysed the sponge core–microbiome network (the core network). The core microbiome was defined as the subset of OTUs found in sponge hosts, which were (i) present with a relative abundance > 0.01% across the whole dataset and (ii) were present in at least two-thirds of the samples from the species they were associated with. These criteria were defined by Astudillo-García et al.[31] and are robust to different filtering criteria. We acknowledge however that this procedure implies a space for time substitution for the presence of microbes, and thus 'transient' microbes are actually transient across species samples and not through time. Filtering according to the core microbiome reduced the number of OTUs to 10,941 resulting in a sponge core–microbiome network containing only 51,207 links.

**Phylogenetic analysis and UniFrac**. To assess the extent to which observed patterns in network modularity were driven by microbial or host phylogeny, phylogenetic trees were constructed for both the sponges and the microbial OTUs in the network. OTU phylogeny was assessed using 16S rRNA gene neighbour-joining phylogenetic construction within the software MEGA7[66] for the subset of OTUs in the core network, while FastTree[67] was used to construct the OTU phylogeny for the global network. Sponge phylogeny was extracted from the Open Tree of Life[68] (OTL), using the R[69] library rotl[70]. To evaluate whether OTUs or sponges that are closer in the tree of life are found in the same network modules, their relationship was analysed using the unweighted UniFrac[71] algorithm provided by the Mothur[72] software package. Unweighted UniFrac determines the extent (quantified as the phylogenetic distance) to which any of the groups (network modules in our case) into which the different OTUs in the phylogenetic tree have been aggregated have a significantly different phylogenetic composition than the other groups. The UniFrac score is thus the difference (or more intuitively, the distance) between microbial communities in a given module to the rest; or between two modules in pairwise comparisons.

**Functional profiling**. To test whether network modules comprise microbes with distinct functional repertoires, we investigated the functional profile of each OTU and associated that information to their community membership. For the core microbiome, we thus extracted functional profiles based on the Kyoto Encyclopaedia of Genes and Genomes (KEGG) Ontology (KO) using Tax4Fun2[73]. Only OTUs that were matched with a 97% similarity to sequences present in the SILVA database were included. Using this threshold, 994 (7.6%) of the 10,941 total core OTUs could be functionally profiled.

Tax4Fun2 extracts functional profiles calculated using complete genomes available through the KEGG database to generate reference data. Functional profiles are obtained from each genome and are then associated with taxonomic keys (i.e. a particular genus) found in the SILVA reference genome database by using the 16S rRNA gene data from each genome. Using sequences aligned against the SILVA database from the original dataset, we used Tax4Fun2 to check whether a reference profile was available for each taxonomic classification. Tax4Fun2 then calculated a predicted metagenome incorporating the abundance of each OTU. OTUs assigned to a taxonomic key having no functional reference profiles would not be included in the prediction (called Fraction of Unexplained; FTU).

For the present study, we generated the reference data as follows: we obtained more than 10,000 complete genomes available in RefSeq (the NCBI reference sequence database) and inferred a functional profile for each based on KEGG KO. 16S rRNA gene sequences from each genome were extracted. Functional profiles of each genome were normalised by the number of 16S rRNA genes within the genome. Extracted 16S rRNA gene sequences were clustered at 100% similarity. One sequence of each cluster served as reference for the cluster. A functional reference profile for each 16S rRNA cluster was calculated from all genomes affiliated to a cluster based on the 16S rRNA gene results. To predict a metagenome, OTU sequences were aligned to the reference 16S rRNA gene sequences with an identity cut-off of 97% (this threshold can be decreased, but will reduce accuracy). OTUs with a lower identity were not considered in the downstream calculation. The abundance of the remaining OTUs in each sample and the reference functional profiles of the reference sequence to which the user sequences were associated during the alignment are then matched and an artificial metagenome is thus calculated. The obtained profile is subsequently normalised: the sum of all functions in a sample is 1.

Tax4Fun2 is freely available as an R package at: https://sourceforge.net/projects/tax4fun2/.

**Statistical analyses**. All statistical analyses were performed in R[70]. Sample temperatures and depths were classified into groups according to the network module that the corresponding sponge species belong to. Statistical support for differences in temperature and depth distributions across modules was then assessed using Nemeyi tests for multiple comparisons of (mean) rank sums of independent samples using the PMCMR[74] library.

Non-metric multidimensional scaling (NMDS) to obtain the ordination of community similarity was performed using the metaMDS function of the vegan[63]

package with Bray–Curtis as the dissimilarity measure. Fitting of the environmental factors and sponge traits to this ordination was conducted using the envfit function from vegan[63]. $R^2$ (i.e. goodness of fit) was used to assess the ability of the predictor variables to explain the variability of the microbial communities.

Mantel tests were used to assess the correspondence between the Bray–Curtis distance matrix based on microbial community composition and the phylogenetic distance matrix based on sponge hosts. Community distances were calculated using the function vegdist from the vegan[63] library, while phylogenetic distances were calculated using the function dist.ml from the ape[75] library.

Differences between the distributions of the participation coefficient values for HMA and LMA sponges respectively were tested for statistical significance using the two-sample Kolmogorov–Smirnov test provided by the ks.test function in R. Differences between functional profiles of sponges and the identified modules were explored using principal coordinate analysis (PCoA) of Bray–Curtis dissimilarity matrices of the log2-transformed relative abundances of metabolic functions inferred for each sponge species. Pairwise permutational analyses of variance (PERMANOVA) were used to test for significant differences between groups of sponges (according to their module membership). PERMANOVA was also used to test microbial community differences between host type. To quantify the differences in microbial community composition within groups (i.e. host type), we analysed the homogeneity of the groups dispersion (i.e. variance). PCoA was performed using the pcoa function of the ape library, and PERMANOVA tests were performed using the adonis function in vegan[63]. Multiple pairwise comparisons of the PERMANOVA tests were corrected based on the Benjamini–Yekutieli false discovery rate control. Analysis of the homogeneity of variance among groups was performed using the betadisper function in vegan[63].

To identify microbial functions that were significantly different across modules, an indicator (i.e. multi-level pattern) analysis was performed. All modules that were not significantly different to the rest based on PERMANOVA analysis (above) were considered as a single module and the multipatt function from the indicspecies[76] library was applied. The resulting bi-serial coefficients ($R$) of each function/pathway within each network module were corrected for unequal sample size using the function r.g[77]. Obtained $p$-values for the permutation tests were adjusted for multiple testing using the Benjamini–Hochberg correction[78].

Metabolic pathways identified by the indicator analysis to be significantly different (with adjusted $p$-values of permutation tests < 0.05) among modules of the core network were used to illustrate the metabolic differentiation across them. Constrained distance-based redundancy analysis was performed on the log2-transformed relative abundances of the subset of metabolic pathways thus identified. Module membership was used as the constraining factor. This analysis was performed using the dbrda function from vegan[63].

**Reporting summary**. Further information on experimental design is available in the Nature Research Reporting Summary linked to this article.

**Code availability**. Computer code to conduct the network construction and analyses described here can be made available upon request.

## Data availability
The datasets analysed during the current study have been made available previously in ref. [30]. Raw sequence data have been deposited in the European Nucleotide Archive (BioProject: PRJEB18736; Accession number: ERP020690). Quality-filtered, demultiplexed fastq files, and QIIME resulting OTU tables are available at the *Qiita* database: http://qiita.microbio.me (Study ID: 10793). Additional information derived from the sequence data including an OTU abundance matrix, an OTU taxonomic classification table, an OTU representative sequence FASTA file, and the metadata for this dataset, which are the only three data sources used in this study, are available from the *GigaScience* repository, GigaDB (https://doi.org/10.5524/100332)[30]. The authors declare that all other data supporting the findings of this study are available within the article and its Supplementary Information files.

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

## Acknowledgements

M.L. and J.M.M. are supported by the French ANR through LabEx TULIP (ANR-10-LABX-41; ANR-11-IDEX-002-02) by a Region Midi-Pyrénées Project (CNRS 121090), and by the FRAGCLIM Consolidator Grant, funded by the European Research Council under the European Union's Horizon 2020 research and innovation programme (Grant Agreement Number 726176). M.L. acknowledges support from the Embassy of France in Australia through its scientific mobility programme, which benefited this collaboration. T.T. is supported by the Australian Research Council.

## Author contributions

M.L., T.T., N.S.W. and J.M.M. conceived the research. All authors designed the research. M.L. and B.W. analysed the data. M.L., T.T., N.S.W. and J.M.M wrote the paper. All authors commented on the manuscript.

## Additional information

**Competing interests:** The authors declare no competing interests.

