## [Peer Review File · Nature Communications]

Reviewers' Comments:

Reviewer #1:

Remarks to the Author:

Modularity and functioning of the global sponge-microbiome network

Lurgi et al use network analysis to reanalyze data from the Sponge Microbiome Project. They find that the global sponge microbiome can be subdivided into 5 modules consisting of a larger number of host-microbe links than would be expected under a null model of network assembly. Microbes and sampled hosts within these modules were then tested for similarity in several properties including the region, depth and temperature of each sample, and whether the microbial OTUs within them are more phylogenetically clustered than expected. Care was taken to address some surprising results, and 1 module consisting largely of samples from a single location in New Zealand with a very high proportion of Firmicutes was traced back to potential sample contamination. A similar analysis was undertaken on association networks for microbes that were prevalent across 66% of samples within a given sponge species (a species-wise 66% 'core microbiome').

Major comments:

Overall I think this is an important analysis trying to understand whether there are portions of the sponge-microbe association network that we can start to explain in terms of environmental or host factors. I think the potential strength of this mode of analysis is that specific commonalities in OTUs across samples can be identified, whereas these can be harder to find when reduced to average UniFrac distances in standard PCoA plots. A significant challenge for this mode of analysis is trying to relate sometimes abstract network properties back to biological conclusions in a way that is defensible.

I thought the manuscript represented a serious effort and quantified the different network compartments identified from a variety of perspectives. However, I had some reservations about some methods, especially the microbial phylogenetics, and the ways in which network results were used to draw biological conclusions in some places.

My main recommendations for revision are:

-- I'm not normally one to nitpick phylogenetic methods when the main focus is on microbiome comparison. Unfortunately, FastTree phylogenies built de novo on short reads are bad enough that they often become unusable. I think this phylogeny needs to be minimally checked by hand in iTOL, coloring by known phyla, for reasonability (we've seen entire phyla misplaced with an identical procedure). Most likely the bacterial tree needs to be replaced with a phylogeny derived from insertion into a reference phylogeny or topological constrain by a reference phylogeny. I'm sorry to ask for this revision, since it might be some extra work. If all major groups look correctly placed on manual inspection maybe it isn't necessary on this dataset. In the specific comments I've added a citation to a recent paper quantifying the issues with FastTree phylogenies of short read data and suggesting an insertion-based alternative that scales well to large datasets. On the positive side, this might explain why no significant association between host phylogenetic distances and the microbial UniFrac distance matrix was detected in some of the Mantel test results. If so perhaps a better tree will recover that association.

-- I find the observations about greater variability in low microbial abundance (LMA) sponges very interesting. I would love to see this finding further explored in relation to the Earth Microbiome Project

paper's observation of nestedness of the microbes in host-associated microbial communities within free-living microbial communities, and in terms of similarity to reference water/environmental samples. Are LMA sponges more restrictive (in a variety of ways) than HMA sponges? Are HMA sponges closer to environmental microbiomes?

-- I also would love to see some additional exploration, and ideally a direct test, of the idea that water-pumping rate is a key feature driving different microbial abundances in HMA vs. LMA sponges. Is there enough physiological data on this to compare pumping rate vs. richness for a subset of these species?

-- I think a major area where the manuscript could be strengthened would be in more clearly and carefully interpreting the biological meaning of aspects of network structure. There are several places, identified in the minor comments, where I found that the reasoning for how the results obtained ruled out particular hypotheses (e.g. environmental filtering) was not wholly clear to me. In some cases these hypotheses might be read in more than one way, and some additional clarification is needed.

Basically, I'd love to know what the tested alternative biological hypotheses are for each question, and what each would predict in terms of each network properties. A table laying this out would make it much easier to relate findings about e.g. provincial hubs back to the ecological or evolutionary process that they are supposed to test or describe. If some analyses are purely descriptive I think that's OK, but it would be good to convey a more clear-cut sense of what each analysis is supposed to test.

-- Related to this point, many abstract network properties (e.g. 'robustness', 'vulnerability to attack', etc) only make sense in particular contexts. That is, the entities represented by nodes and edges affect whether a particular property can be interpreted at all, and if so what we should take from it. I think that when these properties are described, some carefully considered thoughts on how they relate to fundamental evolutionary and ecological processes would really help readers map the technical aspects of the network analysis to how they should update their understanding of the ecology and evolution of the global sponge microbiome.

I wish the authors all the best in their research, and appreciate the opportunity to read this interesting manuscript.

Specific comments:

--Title: "Modularity and functioning of the global sponge-microbiome network"

-- Does it feel a little awkward to mix adjectives and verbs here? Would 'Modularity and predicted functions of the global sponge-microbiome network' work?

-- Line 21: "Here we present the global marine

22 sponge-microbiome network and reveal a compartmental organisation in both community

23 structure and function. Compartments are linked by a few sponge species that share microbes

24 with other species around the world."

There's a lot of crossover in readership between the sponge and coral microbiome communities. Would 'modules' rather than 'compartments' convey the same idea (modules is already used synonymously in a few places in the ms)? I ask because in many coral papers compartments are used anatomically. If the authors prefer the current usage I definitely don't think there's anything wrong with it, just wanted to raise this point in case it could avoid a little bit of confusion.

-- "Further, we provide evidence that abiotic factors influence the broad structuring of the sponge microbiome but that biotic interactions drive the assembly of more intimately associated 'core' microorganisms."

On a first read I found 'broad structuring' to be a bit ambiguous. Could this be revised to specify which type(s) of broad microbiome structure are mostly biotically driven? For example, if this is about network structure and microbiome structure that could be specified (so casual readers know it isn't about other aspects of broad microbiome structure like richness, evenness, nestedness, etc)

-- Line 31 Tangled webs of myriads of interacting species, as imagined by Darwin¹, are ubiquitous on Earth.

Love this opener and the reference to Darwin's 'tangled bank'. It did seem a tiny bit odd to me to say that complex species interactions were imagined by Darwin, since in many cases he directly observed/documentated them.

-- Line 41. "The diversity and composition of ecological communities is thought to be driven by the combination of two main factors: (i) which species arrive, and (ii) whether those that arrive manage to stay⁷."

I think there needs to be a qualifier here about the time-scales involved. For example, this leaves out the key role of speciation in e.g. neutral theory. If some sponge symbionts are vertically transmitted, speciation is probably important at the scale of the evolution of sponge microbe associations.

-- Line 49 "However, for more intimate associations such as symbioses, one would expect a strong microbial community differentiation to emerge across host species."

Since there is a pretty sizable literature of examples of this now, should some of those examples be cited?

Would it strengthen this point to discuss some of the diverse mechanisms by which host species are thought to end up with different microbes. For example, would it be worth discussing the line of papers about diet vs. vertical inheritance in mammal gut microbiome evolution from the Gordon and Alm labs (as well as others)?

-- Line 56 "For instance, host-associated microbial communities may maintain an ancestral signal of host evolution, which can result in related hosts having more similar microbiomes than phylogenetically divergent ones."

Is it worth mentioning that this pattern is known as phylosymbiosis and citing papers from that literature?

-- In marine sponges, host type (i.e., whether it is capable of harbouring highly abundant microbial communities) has been identified as a good indicator of microbiome diversity^{14,15}.

It makes great sense to talk about HMA vs LMA sponges here, but can readers be clued in here about the state of understanding in the field about the host traits that are hypothesized to drive higher microbial abundance and richness in some sponges? The fundamental observation cited here is that

some sponges have both numerically dense/abundant microbial communities (in terms of cell counts) and high richness (in terms microbiome sequencing studies), and Moitinho-Silva et al., 2017's observation that these microbial communities can be separated compositionally and by machine learning. Are there convergent host traits that are thought to drive these similarities? For example, maybe the reference to different water pumping rates as a potential explanation (from line 80) would be useful here.

-- Marine sponges are suitable models for exploring microbiome assembly in host-associated
70 systems as they can harbour dense and diverse microbial communities¹⁶ whose metabolic
71 functions underpin host health and survival¹⁷.

Since most animal microbiome studies mention 'highly diverse' communities, yet those communities vary by many orders of magnitude in richness, it would be really useful to make this statement more quantitative. For example, McDevitt-Irwin et al. (<https://www.frontiersin.org/articles/10.3389/fmars.2017.00262/full>), recently raised the issue that many coral papers talk about a 'diverse' microbiome without also mentioning that that microbiome was typically less diverse than surrounding seawater (e.g. Fig 1 in that paper). I would be really interested to know where HMA and LMA sponges fall in terms of richness relative to other species sampled in the EMP when sampled with consistent methods and rarified at even depth. I don't think there's a "wrong" answer here that would make sponges less interesting, but if this point is important to make, then I think a bit more quantification would help make it.

-- Several lines of evidence suggest that microbial
72 selection within sponges should be driven by factors beyond stochasticity or environmental
73 filtering.

These hypotheses seem really different, and the evidence presented seems to speak more to stochasticity than environmental filtering. I guess I'm also not totally clear about exactly how environmental filtering is defined in this context, and how it would be falsified. A few extra sentences would go a long way to making the subsequent presentation of evidence more impactful. For example, does the 'environmental filtering' hypothesis mean that over evolutionary time-scales a subset of microbes with compatible traits colonize sponges (filtering by the host environment)? Or is the 'environmental filtering' hypothesis that water-column bacteria are filtered by the environment (i.e. traditional non-host habitat filtering) and that the distribution of these environmental bacteria drives sponge microbiome composition?

-- Line 72 How do these concepts of stochasticity and environmental filtering relate to the observation of nestedness of animal and plant microbial communities within the set of environmental OTUs reported in the Thomson et al., EMP paper?

-- These include scale-free degree distributions and the prevalence of weak
76 intraspecific interactions^{16,19}

I get the point about scale-free degree distributions (we're not connecting random sponges and microbes with uniform probability like in an Erdos-Renyii network), but I don't understand how 'weak interspecific interactions are evidence ****against**** stochasticity or environmental filtering. Does this mean weak microbe-microbe correlations in co-occurrence analysis (if so, weak relative to what?). If there were strong inter-specific interactions, would that be evidence for stochasticity or environmental filtering?

-- The identification of modules (or compartments) in complex networks has revealed not only the 88 structured organisation of these networks into tightly linked communities, but perhaps more 89 importantly, the link between this organisation and their functioning and robustness. For 90 instance, the modular organisation of metabolic networks is known to have implications for 91 their robustness and information processing capabilities²².

These points might be good, but I really think more specific examples are needed if we're supposed to understand what 'function' and 'robustness' mean here. For instance, small-world networks are highly robust to random loss of nodes, but vulnerable to 'attack' (meaning targeted removal of key nodes), while Erdos-Renyii random networks show the opposite property. This makes sense if talking about human social networks in some sort of conflict, or computer systems and information security, but does it actually make sense in terms of host-microbe interactions?

I also think more discussion is needed of how much we can generalize these observations across different sets of entities represented by network nodes and edges. What do the implications of information processing capabilities in other networks mean in the context of host-microbe association? Is there external evidence that makes this inference reasonable? Can it be falsified?

Similarly, the way in which marine 92 food webs emerge from the connection between seemingly isolated communities from the⁹³ pelagic and benthic zones²³, points to the role of compartmentalisation in maintaining food⁹⁴ web structure and robustness to perturbations.

In reference to the above comment, I think this is a great reference. In particular, I thought it was really neat how phylogenetically related sharks occupied distinct network modules in the Rezende et al. paper, suggesting niche differentiation. This raises an interesting point -- do phylogenetically related sponges tend to occupy different network modules as these authors observe in Sharks? If so I guess that might also explain the observation of low concordance between host phylogenies and microbial community dissimilarities in the Mantel test results.

Line 97: Additionally, deciphering the organisation of a global host-microbiome network of ecological interactions can facilitate analysis of connectivity profiles across host species, providing insight into how microbiome assembly influences microbiome structure.

I follow the first part of this sentence, but I am not yet convinced that these results support the second part – how microbiome assembly influences microbiome structure. This is a point where a very clear set of hypotheses about each assembly process and predictions about the network properties it would produce would help greatly. Results sections could then refer back to this table to make it clear to readers which specific findings lead you to make particular inferences about the way sponge microbiomes have assembled

Line 106 -- To examine the structuring of the more intimate interactions between sponges and 106 microbes, we also constructed a sponge-core microbiome network where we restricted our 107 analyses to OTUs that were present in at least 2/3 of the samples and with a relative 108 abundance larger than 0.01% across samples of the sponge species they were associated 109 with²⁶.

Can we change 'more intimate' to 'most prevalent' or equivalent? Higher prevalence associations don't

seem like they are necessarily more intimate than low-prevalence associations. For example, human skin may commonly (core microbiome) contact soil bacteria from wind-dispersed dust, but rarely contact skin pathogens, yet probably the pathogenic interaction is more intimate (in the sense of having physiological effects on the host and potentially reflecting host-microbe coevolutionary history).

We analysed the structure of these complex ecological networks and use phylogenies and
112 environmental data to reveal whether: (i) the global network is organised into compartments
113 in which host species with highly overlapping microbial communities group together,
114 reflecting potential commonalities in the assembly of their microbiomes, (ii) the main drivers
115 of this compartmental organisation are environmental factors, host type, and/or microbial
116 evolution within hosts, and iii) compartments correspond to unique predicted functional
117 profiles.

These seem like pretty clear and testable hypotheses. I love this part! How do the potential outcomes of these tests relate to some of the other goals (e.g. understanding assembly) mentioned in the intro? What would the implications of the alternative outcome of each hypothesis be biologically?

>These further enquiries

150 indicated that sponges from this location have probably been contaminated from external
151 sources (personal communication from the contributors of these samples).

I appreciate the follow-up and candid reporting here. Was it possible to determine what the likely source of the contamination was (sediment, human microbiome etc)?

>124 Modularity analysis of the global sponge-microbiome network (the global network from now
125 on) detected 5 network compartments.

Even though it is described in methods, a few sentences about what this analysis was would be very useful here.

>Additionally, OTUs, which were present in seawater samples from
434 the SMP dataset with relative abundances larger than 0.01%, were removed from sponge
435 samples, since they were likely derived from seawater that was part of the sponge sample.

While I understand the motivation behind this environmental filtering step, does it preclude testing the extent to which local environment drives the microbiome? For example, if 99% of physiologically important OTUs in sponges were derived from the local environment early in development (e.g. not just transiently present during feeding), and 1% were vertically inherited, wouldn't this procedure remove the environmental microbes and leave the vertically inherited ones? Does that limit what kind of inferences we can make about the remaining microbes? I don't have any easy answers for how to deal with this issue in a filter-feeding organism (and I think it's tough in any organism), but I do think it should be discussed.

Line 454: This discussion of the modularity function is illuminating and critical to the paper. As modularity is defined relative to a null model, which null model of network structure was used? Is this just the maximum possible modularity (Q_{max}) or something else?

> Sponge-associated microbial communities can be comprised of different types of
516 microorganisms according to the intimacy and repeatability of the association with their host,

517 forming a continuum from core through transient to opportunistic microbes^{43,50}

In the context of sponge microbiomes, is there evidence that core microbes can't also be opportunistic pathogens under the right conditions? In other organisms I'm used to thinking about prevalence (i.e. being part of the 'core') and mutualism, commensalism, opportunism or pathogenicity as separate measures, that may be partially mutually informative but really represent separate things.

Is there tension between the hypothesis that core microbes are the functionally important ones, and the observation of matching between host and microbial phylogenies for some microbes? For example, if OTUs of microbe X strictly track the host phylogeny, with each speciation in the sponge phylogeny matching one speciation event for microbe X, then no OTU of microbe X would be part of the global core sponge microbiome. I think I partially agree with the point being made here, but I find these kind of inferences to be really tough and dependent on scale. In particular, if core microbiomes are calculated at an OTU or sOTU level, then I don't think it's possible to say that because something is outside the core microbiome it is transient.

Finally, I'm not even sure I really agree that low prevalence implies transience, since that has to do with presence over time in time-series data. While a transient microbe might have low prevalence, a microbe can be sporadically distributed across individuals but not transient in time-series data.

-- . We identified a phylogenetic signal on the modular organisation of OTUs in the global network (Unifrac score = 0.757989, p-value < 0.001 and see Supplementary Table 3 for full pairwise comparisons), supporting the notion that specific host-associated microbial communities have evolved within (or in association with) groups of sponge species that harbour phylogenetically similar microbiomes.

Neat finding, which complements the observation of concordance between host phylogeny and microbiome composition in the previous Sponge Microbiome Project paper.

Just to clarify, is 0.757 the the ****difference**** in UniFrac score between within vs. between module comparisons?

-- Line 183 This suggests that this genus of LMA sponges possesses a highly diverse microbiome that is distinct across its constituent species, while at the same time being shared with many other sponges from different phylogenetic backgrounds.

This seems like a really important point, and I think it could be emphasized and explored more. If HMA sponges are similar in (core) composition whereas LMA sponges are variable, that really might speak to assembly processes, especially in combination with some of the observations about nestedness from the EMP. Do the authors think this is due to more extreme filtering based on host traits in LMA sponges? Could this be tested by asking if LMA sponges differ more from environmental samples than HMA sponges? I'd also love to hear the authors thoughts on how this relates to the hypothesis that LMA vs. HMA sponges differ in microbial abundance based on water-pumping rates.

Line 192 This suggests that HMA sponges are more similar in terms of microbial community composition than LMA sponges, which can harbour microbes that are shared across many different sponge species.

Since all the data is available, this should be directly tested using e.g. Adonis on the existing unweighted UniFrac distance matrix, treating LMA vs. HMA status as the categorical variable (or reference past results where this was already done).

Line 191 In contrast, LMA species showed greater variability in the fraction of links contained within their compartment, with generally high participation coefficients ($P > 0.4$) (Fig. 4).

I must be missing something. In Fig 4, not all LMA sponges show participation coefficients > 0.4 . It looks like about 13 LMA (blue) species fall below this threshold.

OTU phylogeny was assessed using 16S rRNA gene
535 neighbour-joining phylogenetic construction within the software MEGA760 for the subset of 536 OTUs in the core network, while FastTree61 was used to construct the OTU phylogeny for the 537 global network.

Was this a de novo phylogeny? While I know this was (very unfortunately) default in QIIME for some time, de novo phylogenetic inference on short read fragments can give surprisingly bad results. This has been quantified recently by Janssen et al., (<http://msystems.asm.org/content/3/3/e00021-18>): "However, short (e.g., 150-nucleotide [nt]) DNA sequence fragments do not contain sufficient phylogenetic signal to reproduce a reasonable tree, introducing a barrier in the utilization of critical phylogenetically aware metrics such as Faith's PD or UniFrac." I normally don't fuss too much about slightly sub-optimal phylogenetic methods, but I'd really urge the authors to pay attention to this one.

I'm trying to find a way to avoid asking the authors to redo this with something like SEPP, since I know these downstream analyses can be expensive to run. One quick way to check whether this is strictly necessary or if, despite some misplacements, the results are 'good enough' for the intended purpose is to manually inspect the placement of key taxa. For example, it's pretty quick to plot the inferred microbial tree with Greengenes taxonomy labels in a tool like iTOL. Maybe de novo FastTree inference worked great on this dataset, but in our (anecdotal) experience the results of this quick check on similar data with I think an identical procedure were horrific (e.g. misplacement of whole phyla with well-understood relationships).

If a reanalysis is necessary, the best alternatives that I'm aware of are using insertion methods (pplacer) or topological constrain in RAXML (to match a long-read based phlogeny like SILVA or Greengenes). The Janssen paper describes an insertion workflow suitable for large datasets.

On the upside, this might rescue the association between bacterial phylogeny and host phylogeny in Mantel tests (if important bacterial groups are badly misplaced then real signal might get swamped).

> This suggests that, even though evolutionary processes play an important
220 role in structuring microbial communities within compartments, sponges are able to acquire
221 and maintain microbes that are not necessarily phylogenetically related to other members of
222 their associated microbial community.

Do these non-phylogenetically structured provincial hub OTUs show a correspondingly stronger response to geography? If so they might be due to regional/ sampling location effects (i.e. unrelated sponges in the same or nearby spots take up environmental microbes, which are therefore 'provincial hubs').

Together with

258 the strong phylogenetic signal detected for the observed compartmentalisation, this suggests
259 that different sets of LMA sponges (and to a lesser extent HMA species given their presence
260 in other compartments) have different functional potential by harbouring specific and
261 distinctly structured bacterial communities.

Sure – if bacterial communities differ then functional profiles predicted based on those bacteria will almost always differ as well. (I know it's technically possible that they wouldn't in extreme cases of functional convergence, but I don't think this has ever been reported). Is it possible to compare the variance in microbiome composition explained by a distance matrix constructed for the functional profile vs. the OTU table in e.g. PERMANOVA or Adonis? If compartments explained more of the variance in predicted functional profile then it would be a stronger clue that there is an unusually strong effect from specific functions (e.g. perhaps due to habitat filtering on traits like biosynthesis of secondary metabolites across diverse lineages rather than just the predicted similarities in some categories based on microbial relatedness). I'd also be interested to see the same for the table of just biosynthesis of secondary metabolites.

Our results show how abiotic factors can

276 have a strong influence on the broad structuring of host-associated microbial communities,
277 but a common microbial evolutionary origin is most likely responsible for the assembly of
278 core, likely symbiotic, microbial communities.
279

Is this statement defensible given the results? If there was a common origin for all (or most or many) of the microbes in the core microbiome of sponges, and modern sponge communities mostly descend from that ancestral set of microbial symbioses, then shouldn't we expect that host phylogeny would significantly correlate with microbiome community similarity?

Let's say that phylogenetic re-analysis shows associations between microbiome structure and host phylogeny. I'm still not sure these results rule out alternative hypothesis for the evolution of microbial associations in sponges. As I understand it, the core microbiome was defined based on consistency within each sponge species. Is that right? If so, then it seems like if there were several events in which environmental microbes were acquired and subsequently vertically transmitted within specific sponge lineages, we could see core microbiomes with members that are not derived from common inheritance with the LCA of sponges, yet which still show significant phylogenetic signal.

Reviewer #2:

Remarks to the Author:

Summary

=====

The authors constructed a bipartite host-OTU network from a sponge microbiome dataset collected previously and used it to assess the role of host phylogeny, host ecotypes, environmental factors and functions in shaping the microbial community structure. The analysis is of interest and has been carried out carefully, so that I have only a few major comments.

Major

====

The problem with network construction from presence/absence data is the dependency on the

sequencing depth. If the data set had been sequenced deeper, OTUs may have been found on hosts where they are now deemed absent, and vice versa for a more shallowly sequenced data set. So how robust are the main findings to a difference in sequencing depth? Do the main conclusions still hold if the network is constructed from 16S data that are rarefied to a different depth?

Are some microbial phyla/classes significantly enriched/depleted in some modules?

The authors could make full use of their abundance data and look for significant positive/negative associations between microbial taxa across hosts. For instance, it would be interesting to find a cluster of taxa that are enriched in the pathways for secondary metabolite production that distinguished one of the modules. Such an abundance-based network analysis could also take environmental factors into account to find out whether sponge-associated microbial taxa tend to co-vary with other environmental factors besides temperature.

Minor

=====

Is there some way to assess sponge age? May its age affect the diversity of the microbiota that a sponge hosts?

Were some sponge species sampled more than once? If yes, was the microbial community conserved within one species?

"This is consistent with recent findings by Björk et al., of prevalent ammensal and commensal interactions between microbes in sponge-associated microbial communities"

Did these authors validate microbial interactions experimentally? If these are only predicted links, there is no guarantee that they represent ecological interactions, since they could be false positives or indirect.

I assume the order of the modules plotted in Figure 1 a-c (and Supplementary Figure 4 a-c) is the same, but it is not explicitly stated. In addition, how about a Figure 1d (Supplementary Figure 4d) that would show the major microbial phyla or classes present (or significantly enriched/depleted) in each module?

The second cluster in Figure 1b is the only one with an even mixture of HMA and LMA sponges. Is there a hypothesis of what could have overridden the host ecotype as a shaping factor behind community structure?

Please explain briefly how Tax4Fun2 obtains relative abundances of functions. The OTUs were only resolved to species level, so how was possible functional variation on strain level accounted for?

Supplementary Figure 2: I do not see the black or red dots mentioned in the caption.

Supplementary Figure 5: Some pathways are strange, for instance there is a pathway called colorectal cancer and another called influenza A. In this context: is there an ecological hypothesis as to why some secondary metabolite pathways are more abundant in some modules than in others?

L. 178: seem -> seems

L. 402: ammensal -> amensal

Responses to the reviewers' comments:

In the following we give a point-by-point response to the comments by the reviewers on our manuscript. All our responses are presented in orange. Line numbers cited next to the changed sentences correspond to the revised version of the manuscript without tracked changes.

Reviewer #1 (Remarks to the Author):

Modularity and functioning of the global sponge-microbiome network

Lurgi et al use network analysis to reanalyze data from the Sponge Microbiome Project. They find that the global sponge microbiome can be subdivided into 5 modules consisting of a larger number of host-microbe links than would be expected under a null model of network assembly. Microbes and sampled hosts within these modules were then tested for similarity in several properties including the region, depth and temperature of each sample, and whether the microbial OTUs within them are more phylogenetically clustered than expected. Care was taken to address some surprising results, and 1 module consisting largely of samples from a single location in New Zealand with a very high proportion of Firmicutes was traced back to potential sample contamination. A similar analysis was undertaken on association networks for microbes that were prevalent across 66% of samples within a given sponge species (a species-wise 66% 'core microbiome').

Major comments:

Overall I think this is an important analysis trying to understand whether there are portions of the sponge-microbe association network that we can start to explain in terms of environmental or host factors. I think the potential strength of this mode of analysis is that specific commonalities in OTUs across samples can be identified, whereas these can be harder to find when reduced to average UniFrac distances in standard PCoA plots. A significant challenge for this mode of analysis is trying to relate sometimes abstract network properties back to biological conclusions in a way that is defensible.

I thought the manuscript represented a serious effort and quantified the different network compartments identified from a variety of perspectives. However, I had some reservations about some methods, especially the microbial phylogenetics, and the ways in which network results were used to draw biological conclusions in some places.

My main recommendations for revision are:

-- I'm not normally one to nitpick phylogenetic methods when the main focus is on microbiome comparison. Unfortunately, FastTree phylogenies built de novo on short reads are bad enough that they often become unusable. I think this phylogeny needs to be minimally checked by hand in iTOL, coloring by known phyla, for reasonability (we've seen entire phyla misplaced with an identical procedure). Most likely the bacterial tree needs to be replaced with a phylogeny derived from insertion into a reference phylogeny or topological constrain by a reference phylogeny. I'm sorry to ask for this revision, since it might be some extra work. If all major groups look correctly placed on manual inspection maybe it isn't necessary on this dataset. In the specific comments I've added a citation to a recent paper quantifying the issues with FastTree phylogenies of short read data and suggesting an insertion-based alternative that scales well to large datasets. On the positive side, this might explain why no significant association between host phylogenetic distances and the microbial UniFrac distance matrix was detected in some of the Mantel test results. If so perhaps a better tree will recover that association.

We thank the reviewer for this comment regarding the microbial phylogenetics performed in this work. We have now performed the additional analyses suggested by the reviewer and, indeed, the colouring of the tree by known phyla shows some misplacement of sequences (see Figure R1 in this response below). This can result from two causes: an error in the taxonomic classification, due to either the classification method used and/or misclassification of sequences within public databases, which has been well documented in the literature^{1,2}; or an error in the tree-building itself.

Our initial attempt at solving this potential problem in our tree was to reclassify the OTU using pplacer (as suggested by the reviewer) and the latest SILVA release (SSURef version 132). It should be noted that this does not solve the problem of already misclassified sequences in the reference database (i.e., SILVA). We generated the reference data for pplacer as follows:

1.- We obtained the latest SILVA release (SSURef version 132) and clustered sequences with uclust (the default algorithm used by SILVA to generate a non-redundant version of their database) at 97% identity. We chose a 97% cut-off threshold to reduce the size of the full dataset (making it more manageable) but at the same time including enough sequences to produce a reliable classification/placement.

2.- We extracted the aligned sequences for all centroid sequences obtained by the uclust analysis from the aligned version of the SILVA database to generate the reference alignment needed by pplacer.

3.- We then attempted to build a microbial phylogeny from this reference alignment using RAxML. This attempt to use RAxML to build the tree failed due to the size of this (reduced) alignment. Only FastTree with double precision (FastTreeDbI; please see <http://darlinglab.org/blog/2015/03/23/not-so-fast-fasttree.html> for details) could calculate a reference tree using the information extracted from SILVA. The computation of this tree took 4 weeks on a single core @ 2GHz and 2TB of RAM memory. Available hardware is clearly here a limiting factor for this computation and does not allow us to produce a reference tree with RAxML for pplacer analysis.

This relates back to the second possible cause of errors in the construction of the microbial phylogeny: errors in the tree building process. To assess the extent to which the tree construction procedure affected our results, we compared the two algorithms introduced above for building phylogenies: FastTree (the one we used) vs RAxML (the one suggested by the reviewer).

It is important to note that FastTree has been shown to reconstruct reliable trees when compared to well-established tools like RAxML³. Even though we appreciate that short sequences can be problematic in phylogenetics, we are not able to successfully complete a RAxML analysis on our complete dataset or the reference alignment (see above), even with having access to a substantial high-performance/high-memory node. RAxML did simply not finish the computation of the whole tree even after several attempts.

Nonetheless, in order to further address the reviewer's concern, and assess the differences between tree building algorithms for our dataset, we generated 1000 randomly sampled subsets of our data (each containing 1000 sequences) and compared trees generated on these datasets using both RAxML and FastTree. We first compared the trees generated for each subset using their cophenetic distances using the *cophenetic* function in R and found high congruence between the two approaches, as indicated by a significant 80% correlation in a Mantel test. Although this result indicates that our phylogenetic inference with FastTree has accuracy consistent with certain standards in the field (i.e., RAxML), it might be argued that there is still a chance that some sequences were misplaced, which could severely impact the subsequent UniFrac analyses.

To evaluate this possibility, we obtained an OTU table consisting of 100 randomly generated samples (using the *sample* function in R) for each of the 1000 subsets for which we had built the corresponding phylogenetic trees (see above). UniFrac distances obtained from both trees of each subset (FastTree and RAxML) were compared using Mantel tests. These tests yielded, on average, a 93% correlation between UniFrac distances calculated from the two tree-calculation methods, clearly demonstrating that UniFrac distances calculated using FastTree phylogenies reflect those generated by RAxML, at least for the dataset used in this work.

We are therefore confident that the inferences and conclusions of our analyses are not biased by the use of FastTree, which was the only method that could handle our large dataset.

-- I find the observations about greater variability in low microbial abundance (LMA) sponges very interesting. I would love to see this finding further explored in relation to the Earth Microbiome Project paper's observation of nestedness of the microbes in host-associated microbial communities within free-living microbial communities, and in terms of similarity to reference water/environmental samples. Are LMA sponges more restrictive (in a variety of ways) than HMA sponges? Are HMA sponges closer to environmental microbiomes?

We agree with the reviewer that exploring our findings in light of results presented in the EMP paper would be interesting. However, direct comparison with the figures/data presented in that earlier work are likely confounded by the different ways in which the data were processed in both studies (e.g., rarefaction thresholds, OTU filtering). Effective comparison of both datasets would involve re-processing both datasets under a common protocol and thus remains beyond the scope of this study. However, on this point, the nestedness of microbes in sponge-associated microbial communities within free-living microbial communities is a pattern unlikely to emerge unless the classification used for OTUs is very broad (e.g. phylum-level). As shown by Thomas et al.⁴, microbial communities from seawater are compositionally very different to both HMA- and LMA- microbial communities (Figure 5 in Thomas et al.⁴), although LMA sponges can harbour microbial communities more similar to seawater microbiomes than HMA sponges⁵. Additionally, the number of unique OTUs found across sponge species is more than three times larger than that found in seawater (Figure 4b in Thomas et al.⁴), indicating that nestedness of host-associated microbial communities within seawater-derived ones is unlikely, at least at this level of taxonomic resolution.

-- I also would love to see some additional exploration, and ideally a direct test, of the idea that water-pumping rate is a key feature driving different microbial abundances in HMA vs. LMA sponges. Is there enough physiological data on this to compare pumping rate vs. richness for a subset of these species?

We agree with the referee that this would be an informative test for understanding the ecology of HMA vs LMA associations, however there is currently insufficient data on pumping rates of sponges to support this analysis.

To acknowledge the importance of this hypothesis however, we have added the following text in the discussion of the manuscript to highlight the potential role of water pumping rates in sponge microbiome assembly: 'We hypothesise that the greater differentiation observed across LMA sponges compared to HMAs is due to physiological traits related to the sponges' ability to filter seawater, such as water pumping rates^{6,7}, however, insufficient physiological data is currently available to test hypotheses of this kind. This therefore, constitutes an important area for future research...' (lines 324-328)

-- I think a major area where the manuscript could be strengthened would be in more clearly and carefully interpreting the biological meaning of aspects of network structure. There are several places, identified in the minor comments, where I found that the reasoning for how the results obtained ruled out particular hypotheses (e.g. environmental filtering) was not wholly clear to me. In some cases these hypotheses might be read in more than one way, and some additional clarification is needed.

Basically, I'd love to know what the tested alternative biological hypotheses are for each question, and what each would predict in terms of each network properties. A table laying this out would make it much easier to relate findings about e.g. provincial hubs back to the ecological or evolutionary process that they are supposed to test or describe. If some analyses are purely descriptive I think that's OK, but it would be good to convey a more clear-cut sense of what each analysis is supposed to test.

We agree with the reviewer that a more explicit statement of the hypotheses and how these were tested would improve the clarity of the paper and better convey the implications of the work's findings. We have added this additional information as new paragraphs within the introduction, because a table format would have necessitated considerable repetition of information. In responses to specific comments below we specify the places within the manuscript where clarifications of specific hypotheses are now included.

-- Related to this point, many abstract network properties (e.g. 'robustness', 'vulnerability to attack', etc) only make sense in particular contexts. That is, the entities represented by nodes and edges affect whether a particular property can be interpreted at all, and if so what we should take from it. I think that when these properties are described, some carefully considered thoughts on how they relate to fundamental evolutionary and ecological processes would really help readers map the technical aspects of the network analysis to how they should update their understanding of the ecology and evolution of the global sponge microbiome.

We cannot agree more here. Many network properties do not make sense/do not apply in particular contexts, and they may not have the same meaning for different networks. We have now revised our presentation of the interpretation of network properties based on the ecological and evolutionary aspects of the system. In particular, we modified the section of the introduction that presents the network-related ideas: 'One of the key structural features of complex networks is their modularity⁸. The identification of modules in complex networks has revealed not only the structured organisation of these networks into tightly linked communities, but perhaps more importantly, the link between this organisation and their functioning and robustness. For instance, the modular organisation of metabolic networks is known to have implications for information processing capabilities (i.e., sharing metabolic products across different pathways), and their robustness against the failure of specific metabolic reactions⁹. Similarly, the emergence of marine food webs from the connection between seemingly isolated communities from the pelagic and benthic zones¹⁰, points to the role of modularity in maintaining food web structure and robustness to perturbations such as species loss. In sponge-microbiome networks, a modular structure can reveal common metabolic pathways across host species (i.e., equivalent information processing capabilities). This can in turn facilitate the understanding of metabolic collapse in given species by looking at other species in the same module (i.e., robustness). A better understanding of the modularity of microbial-host interaction networks is thus essential to uncover fundamental patterns in their structure and function. Additionally, deciphering the organisation of a global host-microbiome network of ecological interactions can facilitate analysis of connectivity profiles across host species¹¹, providing insight into how microbiome assembly influences microbiome structure. For instance, if HMA sponges tend to be more connected within their module relative to LMA sponges, this would suggest that the microbiome assembly processes in HMA sponges affect the structure of the network in predictable ways. On the other hand, if connectivity profiles are homogeneously distributed across hosts, this would provide evidence to support the idea that microbiome assembly processes do not affect host-microbiome network structure.' (lines 98-121)

I wish the authors all the best in their research, and appreciate the opportunity to read this interesting manuscript.

Specific comments:

--Title: "Modularity and functioning of the global sponge-microbiome network"

-- Does it feel a little awkward to mix adjectives and verbs here? Would 'Modularity and predicted functions of the global sponge-microbiome network' work?

Changed to the suggested title.

-- Line 21: "Here we present the global marine sponge-microbiome network and reveal a compartmental

organisation in both community structure and function. Compartments are linked by a few sponge species that share microbes with other species around the world.”

There’s a lot of crossover in readership between the sponge and coral microbiome communities. Would ‘modules’ rather than ‘compartments’ convey the same idea (modules is already used synonymously in a few places in the ms)? I ask because in many coral papers compartments are used anatomically. If the authors prefer the current usage I definitely don’t think there’s anything wrong with it, just wanted to raise this point in case it could avoid a little bit of confusion.

As suggested by the reviewer, and to avoid reader confusion we have changed the use of ‘compartments’ to ‘modules’ throughout the manuscript.

-- “Further, we provide evidence that abiotic factors influence the broad structuring of the sponge microbiome but that biotic interactions drive the assembly of more intimately associated ‘core’ microorganisms.”

On a first read I found ‘broad structuring’ to be a bit ambiguous. Could this be revised to specify which type(s) of broad microbiome structure are mostly biotically driven? For example, if this is about network structure and microbiome structure that could be specified (so casual readers know it isn’t about other aspects of broad microbiome structure like richness, evenness, nestedness, etc)

To avoid confusion, we now define what we mean by ‘broad structuring’ and ‘broad microbiome’, changing the sentence to: ‘Further, we provide evidence that abiotic factors influence the structuring of the sponge microbiome when considering all microbes present, but biotic interactions drive the assembly of more intimately associated ‘core’ microorganisms.’ (lines 25-27).

-- Line 31 Tangled webs of myriads of interacting species, as imagined by Darwin, are ubiquitous on Earth.

Love this opener and the reference to Darwin’s ‘tangled bank’. It did seem a tiny bit odd to me to say that complex species interactions were imagined by Darwin, since in many cases he directly observed/documentated them.

Changed to ‘imagined and observed’.

-- Line 41. “The diversity and composition of ecological communities is thought to be driven by the combination of two main factors: (i) which species arrive, and (ii) whether those that arrive manage to stay.”

I think there needs to be a qualifier here about the time-scales involved. For example, this leaves out the key role of speciation in e.g. neutral theory. If some sponge symbionts are vertically transmitted, speciation is probably important at the scale of the evolution of sponge microbe associations.

We have added the following sentence at the end of the paragraph to acknowledge this:

‘Given the timescales involved in the development of these associations and the resulting differentiation, additional evolutionary aspects, such as speciation, are also expected to play a role in their structuring.’ (lines 51-53)

-- Line 49 “However, for more intimate associations such as symbioses, one would expect a strong microbial community differentiation to emerge across host species.”

Since there is a pretty sizable literature of examples of this now, should some of those examples be cited?

Would it strengthen this point to discuss some of the diverse mechanisms by which host species are thought to end up with different microbes. For example, would it be worth discussing the line of papers about diet vs. vertical inheritance in mammal gut microbiome evolution from the Gordon and Alm labs (as well as others)?

We have added the following sentence to the end of that paragraph to reflect previous work in this field, including appropriate references: ‘In fact, host-associated microbiomes such as those found in the mammalian gut are structured by diverse mechanisms ranging from ‘ecological’ drivers, such as diet^{12,13}, to coevolution¹⁴ and co-speciation¹³.’ (lines 53-55)

-- Line 56 “For instance, host-associated microbial communities may maintain an ancestral signal of host evolution, which can result in related hosts having more similar microbiomes than phylogenetically divergent ones.”

Is it worth mentioning that this pattern is known as phyllosymbiosis and citing papers from that literature?

We have amended this sentence to reflect the existing literature on the subject. It now reads: 'For instance, host-associated microbial communities may maintain an ancestral signal of host evolution, which can result in related hosts having more similar microbiomes than phylogenetically divergent ones, a pattern known as phylosymbiosis¹⁵.' (lines 60-63)

-- In marine sponges, host type (i.e., whether it is capable of harbouring highly abundant microbial communities) has been identified as a good indicator of microbiome diversity.

It makes great sense to talk about HMA vs LMA sponges here, but can readers be clued in here about the state of understanding in the field about the host traits that are hypothesized to drive higher microbial abundance and richness in some sponges? The fundamental observation cited here is that some sponges have both numerically dense/abundant microbial communities (in terms of cell counts) and high richness (in terms of microbiome sequencing studies), and Moitinho-Silva et al., 2017's observation that these microbial communities can be separated compositionally and by machine learning. Are there convergent host traits that are thought to drive these similarities? For example, maybe the reference to different water pumping rates as a potential explanation (from line 80) would be useful here.

Unfortunately, it is not currently known what drives the differences between LMA and HMA sponges. Although there have been some hypotheses raised, including water pumping rates, none of them have yet been shown to drive these differences. Given this current state of understanding, we are reluctant to speculate further and prefer to limit the argument to HMA-LMA dichotomy being a good way to predict microbiome diversity. To acknowledge the importance of a better understanding of this dichotomy to achieve a good understanding of sponge microbiome assembly, we have added this sentence to the discussion: (further research) '...will shed light on the current HMA-LMA dichotomy for which we lack convincing and unequivocal mechanisms.' (lines 328-329)

-- Marine sponges are suitable models for exploring microbiome assembly in host-associated systems as they can harbour dense and diverse microbial communities whose metabolic functions underpin host health and survival.

Since most animal microbiome studies mention 'highly diverse' communities, yet those communities vary by many orders of magnitude in richness, it would be really useful to make this statement more quantitative. For example, McDevitt-Irwine et al. (<https://www.frontiersin.org/articles/10.3389/fmars.2017.00262/full>), recently raised the issue that many coral papers talk about a 'diverse' microbiome without also mentioning that that microbiome was typically less diverse than surrounding seawater (e.g. Fig 1 in that paper). I would be really interested to know where HMA and LMA sponges fall in terms of richness relative to other species sampled in the EMP when sampled with consistent methods and rarified at even depth. I don't think there's a "wrong" answer here that would make sponges less interesting, but if this point is important to make, then I think a bit more quantification would help make it.

To provide a more quantitative assessment of microbial richness and put this richness into the broader context of the EMP, we have now modified the sentence (lines 75-80): 'Marine sponges are suitable models for exploring microbiome assembly in host-associated systems as they can harbour dense and diverse microbial communities⁴ with microbial richness around 103 for both high and low microbial abundance (HMA and LMA) species^{6,16}, consistent with the most microbially diverse environments on the planet¹⁷. Importantly, metabolic functions performed by these microbes underpin host health and survival¹⁸.'

-- Several lines of evidence suggest that microbial selection within sponges should be driven by factors beyond stochasticity or environmental filtering.

These hypotheses seem really different, and the evidence presented seems to speak more to stochasticity than environmental filtering. I guess I'm also not totally clear about exactly how environmental filtering is defined in this context, and how it would be falsified. A few extra sentences would go a long way to making the subsequent presentation of evidence more impactful. For example, does the 'environmental filtering' hypothesis mean that over evolutionary time-scales a subset of microbes with compatible traits colonize sponges (filtering by the host environment)? Or is the 'environmental filtering' hypothesis that water-column bacteria are filtered by the environment (i.e. traditional non-host habitat filtering) and that the distribution of these environmental bacteria drives sponge microbiome composition?

We have modified the sentence to: 'environmental filtering provided by the water column' (lines 81-82) in order to avoid any confusion about the type of environmental filtering we are referring to. We have also added detail about the expectations for the different types of assembly processes so that the reader can better understand what to expect from each of them: 'Under a stochastic scenario such as assembly via water column filtering, none of the microbiome organisation patterns would be expected to emerge because heterogeneous are improbable to emerge stochastically, and different host strategies (i.e., HMA vs LMA) would not be possible if the host-associated microbiome was totally dependent on that of the water column.' (lines 92-96).

-- Line 72 How do these concepts of stochasticity and environmental filtering relate to the observation of

nestedness of animal and plant microbial communities within the set of environmental OTUs reported in the Thomson et al., EMP paper?

Thompson's paper shows that nestedness is minimal at the OTU level (Figure 3c), suggesting that in most cases, microbiome composition is neither stochastic nor determined by the environment. At the phylum level, as expected, microbes should all belong to the same bacterial phyla, but the argument here is that host-associated OTUs are different from those in the environment, not that they will have created their own phyla altogether (although, interestingly, in the case of sponges there is actually some evidence for this). So, in these terms, observations from sponge-associated microbiomes are in line with the data presented by Thompson et al: at the phylum level, there is considerable overlap between environmental and sponge-associated communities, but at the OTU or genus level, there is limited overlap⁴. We have added the following sentence to the discussion to relate sponge microbiome findings back to the general trends found across microbial communities: 'Together with recent evidence on the differentiation of sponge associated bacterial communities from those found in the surrounding environment⁴, this suggests that, as previously found for other host-associated microbiomes; even though at the phylum level there might considerable taxonomic overlap between free-living and sponge-associated microbial communities, at the OTU or genus level, we can expect limited overlap between them¹⁷.' (lines 390-395)

-- These include scale-free degree distributions and the prevalence of weak intraspecific interactions

I get the point about scale-free degree distributions (we're not connecting random sponges and microbes with uniform probability like in an Erdos-Renyii network), but I don't understand how 'weak interspecific interactions are evidence ****against**** stochasticity or environmental filtering. Does this mean weak microbe-microbe correlations in co-occurrence analysis (if so, weak relative to what?). If there were strong inter-specific interactions, would that be evidence for stochasticity or environmental filtering?

The reviewer raises an important point and as we don't discuss interaction strengths within the manuscript, we have removed this reference from the introduction.

-- The identification of modules (or compartments) in complex networks has revealed not only the structured organisation of these networks into tightly linked communities, but perhaps more importantly, the link between this organisation and their functioning and robustness. For instance, the modular organisation of metabolic networks is known to have implications for their robustness and information processing capabilities.

These points might be good, but I really think more specific examples are needed if we're supposed to understand what 'function' and 'robustness' mean here. For instance, small-world networks are highly robust to random loss of nodes, but vulnerable to 'attack' (meaning targeted removal of key nodes), while Erdos-Renyii random networks show the opposite property. This makes sense if talking about human social networks in some sort of conflict, or computer systems and information security, but does it actually make sense in terms of host-microbe interactions?

To more explicitly define what these properties mean in metabolic networks, we have rephrased this example to: 'For instance, the modular organisation of metabolic networks is known to have implications for information processing capabilities (i.e., sharing metabolic products across different pathways), and their robustness against the failure of specific metabolic reactions⁹.' (lines 101-104).

The selected examples were meant to illustrate the sort of properties that network organisation can provide to different systems, without any reference to sponge-microbe networks *per se*. To provide a more explicit link between the examples and the specific case of sponge-microbiome networks, we have now added the following sentence to provide expectations on what modularity means in a sponge holobiont: 'In sponge-microbiome networks, a modular structure can reveal common metabolic pathways across host species (i.e., equivalent information processing capabilities). This can in turn facilitate the understanding of metabolic collapse in given species by looking at other species in the same module (i.e., robustness).'

I also think more discussion is needed of how much we can generalize these observations across different sets of entities represented by network nodes and edges. What do the implications of information processing capabilities in other networks mean in the context of host-microbe association? Is there external evidence that makes this inference reasonable? Can it be falsified?

As mentioned in our response above, these examples were not meant to be directly equivalent to host-microbiome networks, but were rather selected as a way of exposing the reader to the sort of properties that modularity can confer to diverse biological systems. Regardless, we have now tried to make more explicit the relationship of information processing capabilities (and other aspects of modularity) to the sponge-microbiomes networks we study. Please see previous response.

Similarly, the way in which marine food webs emerge from the connection between seemingly isolated

communities from the pelagic and benthic zones, points to the role of compartmentalisation in maintaining food web structure and robustness to perturbations.

In reference to the above comment, I think this is a great reference. In particular, I thought it was really neat how phylogenetically related sharks occupied distinct network modules in the Rezende et al. paper, suggesting niche differentiation. This raises an interesting point -- do phylogenetically related sponges tend to occupy different network modules as these authors observe in Sharks? If so I guess that might also explain the observation of low concordance between host phylogenies and microbial community dissimilarities in the Mantel test results.

Yes, this is certainly a possibility. The niche differentiation hypothesis is one of the possible causes of our observation of phylogenetically related sponges, especially for LMA sponges, belonging to different modules in the network (Figure 2 in our manuscript). However, at present it is not possible to perform metabolic profiling for each sponge separately due to the lack of more comprehensive catalogues for OTU-metabolic correspondences. Hence, whilst this is an interesting hypothesis, which may also explain the low concordance between host phylogeny and microbial community similarities, we do not discuss it in the paper as we are unable to effectively test this hypothesis.

Line 97: Additionally, deciphering the organisation of a global host-microbiome network of ecological interactions can facilitate analysis of connectivity profiles across host species, providing insight into how microbiome assembly influences microbiome structure.

I follow the first part of this sentence, but I am not yet convinced that these results support the second part -- how microbiome assembly influences microbiome structure. This is a point where a very clear set of hypotheses about each assembly process and predictions about the network properties it would produce would help greatly. Results sections could then refer back to this table to make it clear to readers which specific findings lead you to make particular inferences about the way sponge microbiomes have assembled

The rationale here is that if we know the roles of sponges in the network, for e.g., a large fraction of sponges have an intermediate number of links (number of OTUs), while at the same time having most connections within their module; and we further know that all of these are HMA sponge (Figure 3 in the manuscript), then we can hypothesise that the way in which the microbiome in HMA sponges is assembled (e.g., by maintaining some microbial OTUs at high abundance) contributes to the formation of modules within the sponge-microbiome network of very similar microbial communities, ultimately determining their structure. LMA sponges, on the other hand, are in general much more heterogeneous in terms of their connectivity to other modules in the network, indicating that microbiome assembly in LMA sponges tends to make the network more connected in general.

As discussed above, to avoid unnecessary duplication of information in a table format as well as reader confusion from multiple lines of cross-referencing, we have instead revised the introduction to now clearly set out each of the hypotheses. We certainly agree with the reviewer on the value of providing clearer links between the assembly processes and the predictions about network properties. We have therefore revised the manuscript text throughout so that these links are now more explicit.

The main modifications are:

'For instance, if HMA sponges tend to be more connected within their module relative to LMA sponges, this would suggest that the microbiome assembly processes in HMA sponges affect the structure of the network in predictable ways. On the other hand, if connectivity profiles are homogeneously distributed across hosts, this would provide evidence to support the idea that microbiome assembly processes do not affect host-microbiome network structure.' (lines 116-121).

And

'By relating phylogenetic and environmental information to the modular structure of the network we can reveal whether ecological or evolutionary mechanisms (or both) have been involved in the assembly of these complex networks. Additionally, if network modules do not correspond to unique functional profiles, this would suggest that metabolic capabilities of host-associated microbiomes do not influence network structure.' (lines 138-143)

Line 106 -- To examine the structuring of the more intimate interactions between sponges and microbes, we also constructed a sponge-core microbiome network where we restricted our analyses to OTUs that were present in at least 2/3 of the samples and with a relative abundance larger than 0.01% across samples of the sponge species they were associated with.

Can we change 'more intimate' to 'most prevalent' or equivalent? Higher prevalence associations don't seem like they are necessarily more intimate than low-prevalence associations. For example, human skin may commonly (core microbiome) contact soil bacteria from wind-dispersed dust, but rarely contact skin pathogens, yet probably the pathogenic interaction is more intimate (in the sense of having physiological effects on the host and potentially reflecting host-microbe coevolutionary history).

We have changed the terminology here as suggested by the reviewer.

We analysed the structure of these complex ecological networks and use phylogenies and environmental data to reveal whether: (i) the global network is organised into compartments in which host species with highly overlapping microbial communities group together, reflecting potential commonalities in the assembly of their microbiomes, (ii) the main drivers of this compartmental organisation are environmental factors, host type, and/or microbial evolution within hosts, and (iii) compartments correspond to unique predicted functional profiles.

These seem like pretty clear and testable hypotheses. I love this part! How do the potential outcomes of these tests relate to some of the other goals (e.g. understanding assembly) mentioned in the intro? What would the implications of the alternative outcome of each hypothesis be biologically?

As mentioned above, we have now clarified how our hypotheses relate to network structure and also the implications of the alternatives. See response to previous comment for specific amendments.

>These further enquiries indicated that sponges from this location have probably been contaminated from external sources (personal communication from the contributors of these samples).

I appreciate the follow-up and candid reporting here. Was it possible to determine what the likely source of the contamination was (sediment, human microbiome etc)?

Unfortunately, we have not been able to identify the sources of contamination for these samples.

>124 Modularity analysis of the global sponge-microbiome network (the global network from now on) detected 5 network compartments.

Even though it is described in methods, a few sentences about what this analysis was would be very useful here.

We have added the following paragraph to explain how modularity was calculated in the results section: 'To calculate the modularity of the sponge-microbiome network we used the modularity function for bipartite networks (Q) proposed by Barber¹⁹ (see Methods). Q is a measure of the fraction of links found between nodes within the same module to those connecting different modules. This modularity analysis finds an optimal nodes partition into modules that maximises Q.' (lines 148-152)

>Additionally, OTUs, which were present in seawater samples from the SMP dataset with relative abundances larger than 0.01%, were removed from sponge samples, since they were likely derived from seawater that was part of the sponge sample.

While I understand the motivation behind this environmental filtering step, does it preclude testing the extent to which local environment drives the microbiome? For example, if 99% of physiologically important OTUs in sponges were derived from the local environment early in development (e.g. not just transiently present during feeding), and 1% were vertically inherited, wouldn't this procedure remove the environmental microbes and leave the vertically inherited ones? Does that limit what kind of inferences we can make about the remaining microbes? I don't have any easy answers for how to deal with this issue in a filter-feeding organism (and I think it's tough in any organism), but I do think it should be discussed.

The reviewer is correct in identifying this caveat for the work. However, our rationale for performing the environmental filtering step was that OTUs common in the water column are most likely to be found in the seawater within the hosts aquiferous system. Additionally, as previous observations suggest, microbial community composition of seawater is very different to sponge-associated microbial community composition. This suggests that what is common in seawater (> 0.01% relative abundance), is unlikely to form part of the sponge microbiome. To acknowledge this caveat in the paper we have now added the following sentences to the discussion: 'Even though our work represents an important first step towards an improved understanding of causes and consequences of the assembly of the sponge microbiome, deeper insights could be gained if optimised methods for analysing/processing data were developed. For example, in processing the data, we filtered OTUs present in the microbiome by removing those that were common in the surrounding seawater. Whilst there was strong justification for this approach, it also represents a potential caveat, as it is possible that genuine members of the sponge microbiome were removed in the process. This may have caused a slight underestimation of metabolic functions within sponges and modules or affected network connectivity, although the number of OTUs removed by this filtering was negligible compared to the actual size of the complete dataset.' (lines 366-375)

Line 454: This discussion of the modularity function is illuminating and critical to the paper. As modularity is defined relative to a null model, which null model of network structure was used? Is this just the maximum possible modularity (Qmax) or something else?

Yes, it is just the maximum possible modularity. The algorithm used to calculate modularity maximises the Q function. We have now explained this in the main text, as indicated above.

> Sponge-associated microbial communities can be comprised of different types of microorganisms according to the intimacy and repeatability of the association with their host, forming a continuum from core through transient to opportunistic microbes

In the context of sponge microbiomes, is there evidence that core microbes can't also be opportunistic pathogens under the right conditions? In other organisms I'm used to thinking about prevalence (i.e. being part of the 'core') and mutualism, commensalism, opportunism or pathogenicity as separate measures, that may be partially mutually informative but really represent separate things.

At present, there is no evidence for this in marine sponges.

Is there tension between the hypothesis that core microbes are the functionally important ones, and the observation of matching between host and microbial phylogenies for some microbes? For example, if OTUs of microbe X strictly track the host phylogeny, with each speciation in the sponge phylogeny matching one speciation event for microbe X, then no OTU of microbe X would be part of the global core sponge microbiome. I think I partially agree with the point being made here, but I find these kind of inferences to be really tough and dependent on scale. In particular, if core microbiomes are calculated at an OTU or sOTU level, then I don't think it's possible to say that because something is outside the core microbiome it is transient.

Finally, I'm not even sure I really agree that low prevalence implies transience, since that has to do with presence over time in time-series data. While a transient microbe might have low prevalence, a microbe can be sporadically distributed across individuals but not transient in time-series data.

We do not provide evidence for a strict one-to-one matching between host and microbial phylogeny and we agree that if this would occur, the concept of a 'global core microbiome' would be elusive. Our phylogenetic evidence (through the use of Unifrac analysis constrained by network module) is limited to the fact that microbial phylogeny is a good predictor of the modular organisation of the core microbiome. Hence, we do not think there is tension between these arguments and still consider core microbes as the functionally important ones.

We do not fully understand the argument for classifying microbes which do not belong to the core microbiome as 'transient' (i.e., they are not consistently present)? In any case, the rationale employed for the classification and analysis of the core microbiome as presented in this paper has been recently used to characterise the core microbiome within marine sponges (e.g., Astudillo-Garcia et al.²⁰) and is useful for separating between prevalent microbes and those found occasionally within the sponge microbiome.

Even though the term 'transient' can imply a temporal nature, few time series data exist for sponge microbiomes. We consider using a space for time substitution to define the core microbiome a valid approach since core microbes should not only be permanent in time within a given individual, but also consistently found across individuals of the same species. This is especially true if these microbes are symbionts, which is the ultimate membership criterion for the core microbiome. To address the reviewers concern we have added the following sentence to the methods to make clear this distinction: 'We acknowledge however that this procedure implies a space for time substitution for the presence of microbes, and thus 'transient' microbes are actually transient across species and not through time.' (lines 605-608)

-- . We identified a phylogenetic signal on the modular organisation of OTUs in the global network (Unifrac score = 0.757989, p-value < 0.001 and see Supplementary Table 3 for full pairwise comparisons), supporting the notion that specific host-associated microbial communities have evolved within (or in association with) groups of sponge species that harbour phylogenetically similar microbiomes.

Neat finding, which complements the observation of concordance between host phylogeny and microbiome composition in the previous Sponge Microbiome Project paper.

Just to clarify, is 0.757 the the **difference** in UniFrac score between within vs. between module comparisons?

In UniFrac, the score of a tree is a measure of the extent to which any of the modules has a different phylogenetic structure than the rest, i.e., the phylogenetic distance of a group to the rest. This number does not make sense without looking at its significance (the given p-value), which is the actual indicator of whether this distance of a module from the rest is actually significantly different than the random expectation. In any case, it can be said that the UniFrac score is indeed the difference (or more intuitively, the distance) between microbial communities in different modules to the rest (i.e., within vs. between). We have added the following sentence to the methods section to clarify how UniFrac works: 'Unweighted UniFrac determines the extent (quantified as the phylogenetic distance) to which any of the groups (network modules in our case) into which the different OTUs in the phylogenetic tree have been aggregated have a significantly different phylogenetic composition than the

other groups. The UniFrac score is thus the difference (or more intuitively, the distance) between microbial communities in a given module to the rest; or between two modules in pairwise comparisons.' (lines 620-626)

-- Line 183 This suggests that this genus of LMA sponges possesses a highly diverse microbiome that is distinct across its constituent species, while at the same time being shared with many other sponges from different phylogenetic backgrounds.

This seems like a really important point, and I think it could be emphasized and explored more. If HMA sponges are similar in (core) composition whereas LMA sponges are variable, that really might speak to assembly processes, especially in combination with some of the observations about nestedness from the EMP. Do the authors think this is due to more extreme filtering based on host traits in LMA sponges? Could this be tested by asking if LMA sponges differ more from environmental samples than HMA sponges? I'd also love to hear the authors thoughts on how this relates to the hypothesis that LMA vs. HMA sponges differ in microbial abundance based on water-pumping rates.

As discussed above, there is currently insufficient physiological data available for most of the sponge species used in our analyses to test hypotheses about what host traits might correlate with the HMA / LMA divide. To address this point, we have highlighted this as an important area for future research, adding the following sentence to the discussion: 'We hypothesise that the greater differentiation observed across LMA sponges compared to HMAs is due to physiological traits related to the sponges' ability to filter seawater, such as water pumping rates^{6,7}, however, insufficient physiological data is currently available to test hypotheses of this kind. This therefore, constitutes an important area for future research...' (lines 324-328)

Line 192 This suggests that HMA sponges are more similar in terms of microbial community composition than LMA sponges, which can harbour microbes that are shared across many different sponge species.

Since all the data is available, this should be directly tested using e.g. Adonis on the existing unweighted UniFrac distance matrix, treating LMA vs. HMA status as the categorical variable (or reference past results where this was already done).

The reviewer raises a very interesting idea; however, it is unfortunately not possible to test this because the UniFrac distance matrix does not provide information about the distance between sponges, but between modules based on the OTUs within each module. For this reason, we cannot assign host type as the categorical variable to the modules. Also, it is not possible to repeat the UniFrac analysis using sponge species identity as the 'group', because OTUs cannot be uniquely assigned to particular sponges.

We were nonetheless interested in performing a more quantitative test for this statement along the lines suggested by the reviewer. We thus performed the multivariate analysis of variance as suggested (using adonis), using the Bray-Curtis distance matrix between microbial communities of sponges (i.e., the "network matrix"). While this yielded support for the differences between groups, it was not sufficient for determining which group displayed greater variance. For this reason, we performed an additional multivariate homogeneity of groups dispersion test (using betadisper). This supported our suggestion that HMA were more similar amongst themselves than LMA sponges.

We now report these results and rephrase the sentence above: 'The qualitative result from the network analysis was quantitatively confirmed with microbial community composition data. Multivariate analysis of variance performed on the microbial community distance matrix between hosts revealed significant differences across host types ($F = 19.78$, $p = 0.001$). Further, analyses of the dispersion of variance within groups revealed that microbial communities within HMA sponges are more similar amongst themselves than those found in LMA sponges (average distance to centroid = 0.47 for HMA vs. 0.64 for LMA).' (lines 231-237).

We have also added the corresponding description of these methods to the 'Methods' section.

Line 191 In contrast, LMA species showed greater variability in the fraction of links contained within their compartment, with generally high participation coefficients ($P > 0.4$) (Fig. 4).

I must be missing something. In Fig 4, not all LMA sponges show participation coefficients > 0.4 . It looks like about 13 LMA (blue) species fall below this threshold.

Yes, this is correct, not all LMA sponges have $P > 0.4$. The sentence as written was not intended to give the impression that all of them were, hence use of the term 'generally'. However, to avoid any unnecessary confusion, we have rewritten the sentence as: 'In contrast, LMA species showed greater variability in the fraction of links contained within their module, with a large fraction of these sponges (83.12%) having high participation coefficients ($P > 0.4$) (Fig. 4).' (lines 226-229)

OTU phylogeny was assessed using 16S rRNA gene neighbour-joining phylogenetic construction within the software MEGA for the subset of OTUs in the core network, while FastTree was used to construct the OTU phylogeny for the global network.

Was this a de novo phylogeny? While I know this was (very unfortunately) default in QIIME for some time, de novo phylogenetic inference on short read fragments can give surprisingly bad results. This has been quantified recently by Janssen et al., (<http://msystems.asm.org/content/3/3/e00021-18>): "However, short (e.g., 150-nucleotide [nt]) DNA sequence fragments do not contain sufficient phylogenetic signal to reproduce a reasonable tree, introducing a barrier in the utilization of critical phylogenetically aware metrics such as Faith's PD or UniFrac." I normally don't fuss too much about slightly sub-optimal phylogenetic methods, but I'd really urge the authors to pay attention to this one.

I'm trying to find a way to avoid asking the authors to redo this with something like SEPP, since I know these downstream analyses can be expensive to run. One quick way to check whether this is strictly necessary or if, despite some misplacements, the results are 'good enough' for the intended purpose is to manually inspect the placement of key taxa. For example, it's pretty quick to plot the inferred microbial tree with Greengenes taxonomy labels in a tool like iTOL. Maybe de novo FastTree inference worked great on this dataset, but in our (anecdotal) experience the results of this quick check on similar data with I think an identical procedure were horrific (e.g. misplacement of whole phyla with well-understood relationships).

If a reanalysis is necessary, the best alternatives that I'm aware of are using insertion methods (pplacer) or topological constrain in RAxML (to match a long-read based phlogeny like SILVA or Greengenes). The Janssen paper describes an insertion workflow suitable for large datasets.

On the upside, this might rescue the association between bacterial phylogeny and host phylogeny in Mantel tests (if important bacterial groups are badly misplaced then real signal might get swamped).

This is an important point raised by the reviewer and the suggestions on how to deal with this are extremely helpful. We undertook both reviewer recommendations (see also comment above): Firstly, we manually checked the tree, colouring it by known taxa and indeed observed some misplacement, although no consistent errors, just a few branches that seemed oddly placed (see Figure R1 below). Because of this we then undertook the second recommendation to re-build the tree using RAxML. However, attempts to do this failed (please see the response to the first major comment above detailing the specifics on how this was done and the reasons for failure).

Figure R1. De-novo microbial phylogeny obtained using FastTree coloured based on SILVA taxonomy using a colour gradient (each colour refers to an abundant taxonomic group).

Nonetheless, we agree with the reviewer that a small fragment of the 16S rRNA gene contains only limited phylogenetic information. Hence, we also followed the reviewer's recommendation of using insertion methods with SEPP. Using such a reference-guided approach seems to be a good solution, but has one major drawback. The placement of the sequences is highly dependent on the reference data.

Janssen and co-workers²¹ (the study cited by the reviewer, in which this methodology has been proven accurate), analysed clinical samples using SEPP. Clinical samples are usually well studied with multiple reference sequences available in common databases. In contrast, marine sponges host several microbial taxa that are only found in sponge microbiomes²². To exemplify this, we compared the sequences in our dataset with the reference data (SILVA) available for SEPP and found that 40% have less than 97% similarity to their closest match using a local alignment. For 1.4% of the sequences, the identity was below 90% indicating that these sequences represent new genera, families and even orders. Based on this observation, the phylogenetic placement of sequences, in particular those with similarity below 90%, would be inaccurate. We also aligned the sponge-associated OTUs against the latest SILVA release with similar results, indicating that even the generation of a new reference profile for pplacer or SEPP would still miss several important reference sequences for our dataset. This inaccuracy in sequence placement using these methods would likely impact the UniFrac analyses. For this reason, building a microbial phylogeny using SEPP would not be appropriate for our dataset.

These explorations did not provide a clear answer on the optimal tree, but we are confident given the testing described above and in the response to the first major point, that the tree used here does not contain a large number of errors and the results presented reflect the phylogenetic patterns of the sponge-associated microbiome.

> This suggests that, even though evolutionary processes play an important role in structuring microbial communities within compartments, sponges are able to acquire and maintain microbes that are not necessarily phylogenetically related to other members of their associated microbial community.

Do these non-phylogenetically structured provincial hub OTUs show a correspondingly stronger response to geography? If so they might be due to regional/ sampling location effects (i.e. unrelated sponges in the same or nearby spots take up environmental microbes, which are therefore 'provincial hubs').

Again, an excellent suggestion by the reviewer. We tested for this by looking at the geographic locations of the sponge hosts in which the provincial OTUs were found and which belonged to the same module that the provincial OTU is a member of. Once the number of unique geographic locations (bioregions) per provincial OTU was calculated, we looked at the distribution of this number across all OTUs to assess whether provincial OTUs tended to be found in a single or a small number of locations (see Figure R2 in this response). This test revealed that the majority of provincial OTUs (72.73%) are found in more than 5 bioregions within their module, while only 14.29% are exclusive to one bioregion.

Figure R2. Distribution of the number of bioregions in which provincial OTUs are found.

We have added this information to the results section of the manuscript thus: 'To assess whether the module membership of provincial hub OTUs can be linked to ecoregion, and hence environmental specificity, we investigated their geographical origin. Most of these provincial hubs (72.73%) are found in more than 5 bioregions within their module, while only 14.29% are exclusive to a single bioregion. This suggests that these 'connecting' OTUs might have been acquired and conserved by different sponge lineages regardless of their environment.' (lines 263-268)

Together with the strong phylogenetic signal detected for the observed compartmentalisation, this suggests that different sets of LMA sponges (and to a lesser extent HMA species given their presence in other compartments) have different functional potential by harbouring specific and distinctly structured bacterial communities.

Sure – if bacterial communities differ then functional profiles predicted based on those bacteria will almost always differ as well. (I know it's technically possible that they wouldn't in extreme cases of functional convergence, but I don't think this has ever been reported). Is it possible to compare the variance in microbiome composition explained by a distance matrix constructed for the functional profile vs. the OTU table in e.g. PERMANOVA or Adonis? If compartments explained more of the variance in predicted functional profile then it would be a stronger clue that there is an unusually strong effect from specific functions (e.g. perhaps due to habitat filtering on traits like biosynthesis of secondary metabolites across diverse lineages rather than just the predicted similarities in some categories based on microbial relatedness). I'd also be interested to see the same for the table of just biosynthesis of secondary metabolites.

We performed both of the tests suggested by the reviewer and, as expected (given the modularity analysis grouped sponges with more similar microbiomes together), all pairwise comparisons of variance of microbiome composition are significantly different (see Table R1 below). This differs from the comparisons of metabolic capabilities across modules, in which some of the comparisons are not significant (Table 2 in the manuscript). However, as suspected by the reviewer, some of the comparisons between modules are stronger (i.e., the groups differ more between them) when looking at metabolic capabilities than when looking at microbial composition, especially those involving modules 4 and 7 (comparing the value of the F statistics in Table 2 in the paper with those in the Table R1 below). This suggests that, at least for some modules, there is a stronger differentiation by functionality than would be expected by looking solely at differences in microbial community composition.

Table R1. PERMANOVA results for the distances in microbial community composition between modules in the sponge-core microbiome network.

Pairs	F statistic	R²	p-value	adjusted p-value	significance
1 vs 2	1.792036	0.04741835	0.002	0.007655253	*
1 vs 3	2.909253	0.07882177	0.001	0.004503090	*
1 vs 4	14.992475	0.17234221	0.001	0.004503090	*
1 vs 7	3.159964	0.06700522	0.001	0.004503090	*
1 vs 6	1.846244	0.04519985	0.002	0.007655253	*
1 vs 5	1.89126	0.06122315	0.002	0.007655253	*
2 vs 3	2.575383	0.08423062	0.001	0.004503090	*
2 vs 4	9.610809	0.12710893	0.001	0.004503090	*
2 vs 7	3.57221	0.08592783	0.001	0.004503090	*
2 vs 6	1.961832	0.05611355	0.001	0.004503090	*
2 vs 5	1.880112	0.07556685	0.001	0.004503090	*
3 vs 4	12.922287	0.16799145	0.001	0.004503090	*
3 vs 7	2.821246	0.07267273	0.001	0.004503090	*
3 vs 6	3.34297	0.09734074	0.001	0.004503090	*
3 vs 5	2.830382	0.11877201	0.001	0.004503090	*
4 vs 7	17.821639	0.19408975	0.001	0.004503090	*
4 vs 6	14.120442	0.16987930	0.001	0.004503090	*
4 vs 5	8.614467	0.12740568	0.001	0.004503090	*
7 vs 6	4.204621	0.09301308	0.001	0.004503090	*
7 vs 5	3.350515	0.09753901	0.001	0.004503090	*
6 vs 5	1.975193	0.07060515	0.004	0.014581435	.

For the case of metabolic functions involving the biosynthesis of secondary metabolites, we repeated the PERMANOVA analysis that was performed for the metabolic functions in the manuscript, but filtering by only the functions labelled as biosynthesis. This yielded 64 out of 348 functions.

Results from this analysis are similar to what was observed for the whole set of metabolic functions, although there are some exceptions (see Table R2 below). For instance, comparisons between modules 3 and 6, and 3 and 7 become non-significant when exclusively assessing biosynthesis. However, comparisons involving modules 4 and 7 remain significant, suggesting that what sets these modules apart from the rest is indeed the biosynthesis of secondary metabolites. This is in line with what is already presented in the manuscript (Figure 5 in the manuscript).

Table R2. PERMANOVA results for the distances in metabolic functions involving the biosynthesis of secondary metabolites between modules in the sponge-core microbiome network.

Pairs	F statistic	R²	p-value	adjusted p-value	significance
1 vs 2	1.552451	0.04134086	0.184	0.741350844	
1 vs 3	2.237684	0.06350257	0.096	0.432296656	
1 vs 4	8.456323	0.10510452	0.001	0.008505837	*
1 vs 7	6.104588	0.12431807	0.002	0.015310507	.
1 vs 6	1.284683	0.03189012	0.259	0.944147905	
1 vs 5	1.256222	0.04151946	0.242	0.926285647	
2 vs 3	6.371223	0.19091967	0.004	0.025517511	.
2 vs 4	18.489151	0.21883462	0.001	0.008505837	*
2 vs 7	13.551466	0.26807267	0.001	0.008505837	*
2 vs 6	2.525094	0.07107916	0.037	0.188829581	
2 vs 5	1.820907	0.07336184	0.129	0.548626485	
3 vs 4	17.137843	0.21385455	0.001	0.008505837	*
3 vs 7	2.444569	0.06707637	0.051	0.244011198	
3 vs 6	3.46448	0.10352709	0.014	0.082441189	
3 vs 5	5.952096	0.22934935	0.003	0.020877963	.
4 vs 7	22.661018	0.23688874	0.001	0.008505837	*
4 vs 6	18.567771	0.21203886	0.001	0.008505837	*
4 vs 5	13.966512	0.19140989	0.001	0.008505837	*
7 vs 6	7.802473	0.16322321	0.001	0.008505837	*
7 vs 5	10.631076	0.26164889	0.001	0.008505837	*
6 vs 5	2.433387	0.08558204	0.031	0.16950918	

Conclusions drawn from these additional analyses are interesting, particularly the comparison of microbial community composition with metabolic capabilities. Hence, we have now incorporated the first table into the main manuscript and revised the results text accordingly: 'Although microbiome differentiation is a better systematic predictor of the differences amongst modules than metabolic potential (Supplementary Table 5), in many cases (particularly those involving comparisons with modules 4 and 7), metabolic differentiation is stronger than what would be expected by solely looking at differences in microbiome composition (compare the F statistics of Table 2 and Supplementary Table 5).' (lines 306-310)

The relevant table has now been included in the Supplementary information.

Our results show how abiotic factors can have a strong influence on the broad structuring of host-associated microbial communities, but a common microbial evolutionary origin is most likely responsible for the assembly of core, likely symbiotic, microbial communities.

Is this statement defensible given the results? If there was a common origin for all (or most or many) of the microbes in the core microbiome of sponges, and modern sponge communities mostly descend from that ancestral set of microbial symbioses, then shouldn't we expect that host phylogeny would significantly correlate with microbiome community similarity?

In cases where microbes are acquired from the environment (for instance, they provide a useful function and are therefore taken up by multiple sponge species requiring the same function), these microbes could subsequently evolve within non-phylogenetically related hosts and thus display the phylogenetic signal observed. This wouldn't be evolution from an ancestral community that has always been associated with the same group of sponge species, but instead represent microbial evolution within different hosts. We therefore suggest that sponge-associated microbial communities may not necessarily descend from a unique ancestral set of microbial symbioses.

If the host-associated microbial communities would have descended from a unique common ancestral set of microbial symbioses, then yes, host phylogeny would be expected to significantly correlate with microbiome community similarity. However, this is not what we observed, therefore our results represent compelling evidence against this hypothesis, at least for LMA sponges. For HMA sponges the microbes are indeed much more similar, and hence a common ancestral origin is plausible.

Let's say that phylogenetic re-analysis shows associations between microbiome structure and host phylogeny. I'm still not sure these results rule out alternative hypothesis for the evolution of microbial associations in sponges. As I understand it, the core microbiome was defined based on consistency within each sponge species. Is that right? If so, then it seems like if there were several events in which environmental microbes were acquired and subsequently vertically transmitted within specific sponge lineages, we could see core microbiomes with members that are not derived from common inheritance with the LCA of sponges, yet which still show significant phylogenetic signal.

This is the point we make in the response above and believe is the most likely possibility given our observations. A 'common microbial evolutionary origin' is likely to be responsible for the differences observed.

Reviewer #2 (Remarks to the Author):

Summary

=====

The authors constructed a bipartite host-OTU network from a sponge microbiome dataset collected previously and used it to assess the role of host phylogeny, host ecotypes, environmental factors and functions in shaping the microbial community structure. The analysis is of interest and has been carried out carefully, so that I have only a few major comments.

Major

====

The problem with network construction from presence/absence data is the dependency on the sequencing depth. If the data set had been sequenced deeper, OTUs may have been found on hosts where they are now deemed absent, and vice versa for a more shallowly sequenced data set. So how robust are the main findings to a difference in sequencing depth? Do the main conclusions still hold if the network is constructed from 16S data that are rarefied to a different depth?

The reviewer is correct in saying that network construction is dependent on sequencing depth. However, our goal in this work was to include all OTUs found in the sponges regardless of sequencing depth to gain the most comprehensive picture of the global sponge-microbiome network.

That being said, we want to clarify that the only part of the analyses subject to this potential problem is the network construction procedure, and hence its resulting topology. When defining the core microbiome, and the metabolic functions associated with each module in the network we used normalised abundance data (i.e., relative abundances) and hence there is no bias from sequencing depth, at least for this part of the analyses. Our main conclusions regarding these aspects thus still hold.

Given this, and acknowledging the fact that statistically testing the effects of sequencing depth on the topology of our network would confer greater confidence in our results, we performed pairwise permutational analyses of variance (PERMANOVA) on 100 rarefied datasets of the original OTU table using the same modules detected in our network as an independent variable, to test for 'network module' as a source of variation in microbiome composition. In this way, we can test whether the modular partition obtained for the network (i.e., the main topological feature our analyses focus on) is robust across the rarefied datasets. This would give support not only to our findings being robust to differences in sequencing depth, but at the same time provide confidence to the results obtained by the modularity algorithm.

We rarefied the data to 10^4 reads per sample (i.e., the size of the smallest sample in the dataset) using the rrarefy function of the vegan package. This procedure was repeated independently 100 times to obtain 100 different rarefied instances (i.e., randomisations) of the data. We then analysed each of these rarefied datasets using pairwise PERMANOVA with the adonis function from vegan. P-values were corrected for false discovery rates due to multiple comparisons using the Benjamini-Yekutieli correction. The results of this analysis (average and standard deviation across randomisations for each value reported by the test) are presented in Table R3.

Table R3. PERMANOVA results for the differences in microbiome composition between modules of the global sponge-microbiome network for 100 independent rarefactions of the raw data. Shown are the average and standard deviation of each value across the 100 replicates. All comparisons are significant.

Pairs	F statistic	R ²	p-value	adjusted p-value	significance
1 vs 2	2.17 ± 0.005	0.126 ± 0.0002	0.001 ± 0.0003	0.003 ± 0.0009	*
1 vs 3	5.02 ± 0.009	0.088 ± 0.0001	0.001 ± 0	0.003 ± 0.0002	*
1 vs 4	5.11 ± 0.01	0.203 ± 0.0003	0.001 ± 0	0.003 ± 0.0002	*
1 vs 5	2.43 ± 0.006	0.028 ± 0.00007	0.001 ± 0	0.003 ± 0.0002	*
2 vs 3	2.65 ± 0.006	0.048 ± 0.0001	0.001 ± 0.0003	0.003 ± 0.0008	*
2 vs 4	2.96 ± 0.005	0.124 ± 0.0002	0.001 ± 0	0.003 ± 0.0002	*
2 vs 5	1.44 ± 0.002	0.016 ± 0.00002	0.001 ± 0.0006	0.003 ± 0.0017	*
3 vs 4	4.37 ± 0.01	0.07 ± 0.0001	0.001 ± 0	0.003 ± 0.0002	*
3 vs 5	9.78 ± 0.012	0.074 ± 0.00008	0.001 ± 0	0.003 ± 0.0002	*
4 vs 5	5.49 ± 0.006	0.057 ± 0.00005	0.001 ± 0	0.003 ± 0.0002	*

Results from the pairwise PERMANOVA analysis show that all comparisons between the microbiomes assigned to each module by the modularity algorithm are still significantly compositionally different from each other for all the rarefied datasets (i.e., all data randomisations performed). This evidence supports that our main findings are robust to differences in sequencing depth in the data.

Are some microbial phyla/classes significantly enriched/depleted in some modules?

As suggested by the reviewer here and again below, we have now added information on the taxonomic composition of each module for both the global and the core sponge-microbiome networks. These are now presented in Supplementary Figures 6 and 7. Additionally, we have added the following text to the results section highlighting these results:

'Interesting differences were observed in the taxonomic composition of OTUs across modules of the global network (Supplementary Figure 6). For example, the microbiomes of modules 3 and 4 (those comprised mainly of HMA sponges), contain a higher fraction of unclassified Bacteria than other modules, while module 5 harbours a larger fraction of Alphaproteobacteria (Supplementary Figure 6). This suggests that not only ecologically- but also genetically-related microbes are forming the microbiome within modules.' (lines 182-187).

And

'This is also evident when looking at the taxonomic profile of the microbiome across modules (Supplementary Figure 7), where, as was the case of the global network, the module containing the majority of HMA sponges shows a microbiome dominated by unclassified Bacteria.' (lines 210-213)

We also added the following paragraph to the discussion based on these observations: 'Importantly, regardless of the assembly mechanisms behind its emergence, we found that the microbiomes of HMA-dominated modules harbour a high fraction of Bacteria that are currently unclassified. Further research is needed to identify and

classify these sponge-associated bacterial strains if we are to fully understand not only their structure but also their metabolic potential.' (lines 446-450)

The authors could make full use of their abundance data and look for significant positive/negative associations between microbial taxa across hosts. For instance, it would be interesting to find a cluster of taxa that are enriched in the pathways for secondary metabolite production that distinguished one of the modules. Such an abundance-based network analysis could also take environmental factors into account to find out whether sponge-associated microbial taxa tend to co-vary with other environmental factors besides temperature.

Although this is a very interesting suggestion, which will provide further insights into the sponge microbiome organisation, we believe these further analyses are outside the scope of the present work, since they imply the construction and analysis of a different network: an OTU-OTU network, which will exclude the sponge hosts from the network analysis. This suggestion would be better framed in an analysis devoted to understanding the metabolic potential and organisation of the sponge microbiome as a coherent unit abstracted from the hosts where the OTUs have been observed. To acknowledge the interest of this potential research avenue, we have added the following sentence to the discussion section in the manuscript: 'To shed light on the configuration and nature of these microbial interactions within the sponge microbiome, future research should focus on the inference of abundance based OTU-OTU interaction networks, additionally considering environmental factors that might modulate bacterial interactions, in order to reveal the structure of interactions between members of the sponge microbiome²³.' (lines 485-489)

Minor

=====

Is there some way to assess sponge age? May its age affect the diversity of the microbiota that a sponge hosts?

Unfortunately, there is no current mechanism for assessing sponge age, hence no studies have yet assessed microbiome changes with sponge age (with the exception of microbiome variation between adult and larval life history stages).

Were some sponge species sampled more than once? If yes, was the microbial community conserved within one species?

Yes, as stated in the methods, we selected sponge species for which at least three samples were available. Microbial communities are generally conserved within individuals of the same species (as shown by Thomas et al.⁴).

"This is consistent with recent findings by Björk et al., of prevalent ammensal and commensal interactions between microbes in sponge-associated microbial communities"

Did these authors validate microbial interactions experimentally? If these are only predicted links, there is no guarantee that they represent ecological interactions, since they could be false positives or indirect.

In the present work, we do not consider interactions between microbes which is well beyond the scope of the study. The suggested analysis could be done for specific species in a potential future OTU-OTU network analysis as suggested above.

I assume the order of the modules plotted in Figure 1 a-c (and Supplementary Figure 4 a-c) is the same, but it is not explicitly stated. In addition, how about a Figure 1d (Supplementary Figure 4d) that would show the major microbial phyla or classes present (or significantly enriched/depleted) in each module?

The numbers of the modules are already shown on the x axis of panel A. As placing the numbers below panel C would result in the legend for the pie chart being oddly placed, we consider this the optimal location for the module labels.

As mentioned above, we have now added taxonomic profiles of the microbiome across modules in both networks to illustrate the enrichment/depletion of taxa (phyla in this case) within each module. We thought however, that the amount of information presented by the taxonomic profiles was too large to be incorporated into Figure 1 and Supplementary Figure 4 as suggested, so instead added two new figures to the supplementary information (Supplementary Figures 6 and 7) which present this information. We added references to both figures as mentioned in the second major comment above.

The second cluster in Figure 1b is the only one with an even mixture of HMA and LMA sponges. Is there a hypothesis of what could have overridden the host ecotype as a shaping factor behind community structure?

Even though this is an interesting observation, as highlighted in the results section, we do not have a clear hypothesis as to why this was observed, except for the common microbial phylogeny.

Please explain briefly how Tax4Fun2 obtains relative abundances of functions. The OTUs were only resolved to species level, so how was possible functional variation on strain level accounted for?

Tax4Fun2 works in a similar way to its precursor, Tax4Fun, using functional profiles calculated based on complete genomes. Tax4Fun used around 4000 genomes available through KEGG to generate its reference data. Functional profiles are obtained from each genome and are then associated with taxonomic keys (i.e., genera) in SILVA (using the 16S rRNA data of each genome). The user aligns sequences against the SILVA database, (e.g., using BLAST) and Tax4Fun checks whether a reference profile is available for a certain taxonomic classification and would then calculate a metagenome incorporating the abundance of each OTU. OTUs assigned to a taxonomic key having no functional reference profiles would not be included in the metagenome prediction (called Fraction of Unexplained; FTU).

This way of predicting functional profiles still forms the basis of functional prediction in Tax4Fun2, but due to the high heterogeneity at the 16S rRNA level within a genus, a new way to make predictions (here referred to as reference-based prediction) has been included. For the present study, we generated the reference data as follows: more than 10,000 complete genomes available in RefSeq were included and we calculated a functional profile for each of the genomes (KEGG KO). In addition to this, we extracted the 16S rRNA sequences from each genome. Functional profiles of each genome were normalised by the number of 16S rRNA genes within the genome. Extracted 16S rRNA genes sequences were clustered at 100% similarity. One sequence of each cluster served as reference for the cluster. A functional reference profile for each 16S rRNA cluster was calculated from all genomes affiliated to a cluster based on the 16S rRNA results. To predict a metagenome, the OTU sequences are aligned to the reference 16S rRNA sequences with an identity cut-off of 97% (can be decreased but will reduce accuracy). OTUs with a lower identity are not considered in the downstream calculation. The abundance of the remaining OTUs in each sample and the reference profiles of the reference sequence that the user sequences were associated to during the alignment are then incorporated and an artificial metagenome is calculated. The obtained profile is subsequently normalised: the sum of all functions in a sample is 1 (which is in line with the original Tax4Fun).

With regard to the second question, one major limitation of functional predictions in general is that strains of the same species can have different functional profiles. Nonetheless, Tax4Fun and PICRUSt predictions were originally compared to different metagenomes obtained from the same samples. Both tools provided high correlations to the real metagenomes with Tax4Fun displaying higher correlation in most comparisons. The reference-based predictions of Tax4Fun2, which were used here, were compared to the same metagenomes and showed a similar correlation coefficient as seen for Tax4Fun. In addition, neither Tax4Fun nor PICRUSt were tested on marine data. Tax4Fun2 was validated using various marine samples with a Spearman correlation coefficient of 85% indicating a significant similarity between predicted and real functional profiles.

Supplementary Figure 2: I do not see the black or red dots mentioned in the caption.

Fixed.

Supplementary Figure 5: Some pathways are strange, for instance there is a pathway called colorectal cancer and another called influenza A. In this context: is there an ecological hypothesis as to why some secondary metabolite pathways are more abundant in some modules than in others?

Some metabolic pathways retrieved from the functional analysis are outside what we would expect for sponges. However, so as to not arbitrarily remove pathways we considered all results within the analysis.

Unfortunately, we currently lack a clear ecological hypothesis as to why some secondary metabolite pathways are more abundant in some modules.

L. 178: seem -> seems

Fixed

L. 402: ammensal -> amensal

Fixed

References:

1. Edgar, R. C. Accuracy of taxonomy prediction for 16S rRNA and fungal ITS sequences. 1, 1–29 (2018).
2. Edgar, R. Taxonomy annotation and guide tree errors in 16S rRNA databases. (2018). doi:10.7717/peerj.5030
3. Liu, K., Linder, C. R. & Warnow, T. RAxML and FastTree : Comparing Two Methods for Large- Scale

- Maximum Likelihood Phylogeny Estimation. **6**, (2011).
4. Thomas, T. *et al.* Diversity, structure and convergent evolution of the global sponge microbiome. *Nat. Commun.* **7**, 11870 (2016).
 5. Erwin, P. M., Coma, R., Lopez-Sendino, P., Serrano, E. & Ribes, M. Stable symbionts across the HMA-LMA dichotomy: low seasonal and interannual variation in sponge-associated bacteria from taxonomically diverse hosts. *FEMS Microbiol. Ecol.* **91**, 1–11 (2015).
 6. Moitinho-Silva, L. *et al.* Predicting the HMA-LMA status in marine sponges by machine learning. *Front. Microbiol.* **8**, 1–14 (2017).
 7. Weisz, J. B., Lindquist, N. & Martens, C. S. Do associated microbial abundances impact marine demosponge pumping rates and tissue densities? *Oecologia* **155**, 367–376 (2008).
 8. Newman, M. E. J. Modularity and community structure in networks. *Proc. Natl. Acad. Sci.* **103**, 8577–8582 (2006).
 9. Guimerà, R. & Nunes Amaral, L. A. Functional cartography of complex metabolic networks. *Nature* **433**, 895–900 (2005).
 10. Rezende, E. L., Albert, E. M., Fortuna, M. A. & Bascompte, J. Compartments in a marine food web associated with phylogeny, body mass, and habitat structure. *Ecol. Lett.* **12**, 779–788 (2009).
 11. Guimerà, R. & Amaral, L. a N. Cartography of complex networks: modules and universal roles. *J. Stat. Mech.* **2005**, nihpa35573 (2005).
 12. Muegge, B. D. *et al.* Diet Drives Convergence in Gut Microbiome Functions Across Mammalian Phylogeny and Within Humans. *Science (80-.)*. **332**, 970–974 (2011).
 13. Groussin, M. *et al.* Unraveling the processes shaping mammalian gut microbiomes over evolutionary time. *Nat. Commun.* **8**, 1–12 (2017).
 14. Ley, R. E. *et al.* Evolution of Mammals and Their Gut Microbes. *Science (80-.)*. **320**, 1647–1651 (2008).
 15. Brooks, A. W., Kohl, K. D., Brucker, R. M., Opstal, E. J. Van & Bordenstein, R. Phylosymbiosis : Relationships and Functional Effects of Microbial Communities across Host Evolutionary History. *PLoS Biol.* **14**, 1–29 (2016).
 16. McDevitt-Irwin, J. M., Baum, J. K., Garren, M. & Thurber, R. L. V. Responses of Coral-Associated Bacterial Communities to Local and Global Stressors. *Front. Mar. Sci.* **4**, 1–16 (2017).
 17. Thompson, L. R. *et al.* A communal catalogue reveals Earth’s multiscale microbial diversity. *Nature* **551**, 457–463 (2017).
 18. Webster, N. S. & Thomas, T. The sponge hologenome. *MBio* **7**, 1–14 (2016).
 19. Barber, M. J. Modularity and community detection in bipartite networks. *Phys. Rev. E - Stat. Nonlinear, Soft Matter Phys.* **76**, 1–11 (2007).
 20. Carmen, A. *et al.* Evaluating the core microbiota in complex communities: A systematic investigation. *Environ. Microbiol.* **19**, 1450–1462 (2017).
 21. Janssen, S. *et al.* Phylogenetic Placement of Exact Amplicon Sequences Improves Associations with Clinical Information. *Ecol. Evol. Sci.* **3**, 1–14 (2018).
 22. Webster, N. S. & Taylor, M. W. Marine sponges and their microbial symbionts: Love and other relationships. *Environ. Microbiol.* **14**, 335–346 (2012).
 23. Faust, K. & Raes, J. Microbial interactions: From networks to models. *Nat. Rev. Microbiol.* **10**, 538–550 (2012).

Reviewers' Comments:

Reviewer #1:

Remarks to the Author:

I have carefully read the authors responses to my comments. I feel that the authors have taken sensible steps to address each point raised in my initial review, including checking the phylogenetic analysis, doing supplementary analyses on 'provincial hub' OTUs, etc. These supplementary analyses, along with some changes to the manuscript text, have fully addressed my concerns.

I wish the authors best of luck with the manuscript and with their future research exploring the sponge microbiome.

Reviewer #2:

Remarks to the Author:

The authors have performed valuable additional analyses, but they have not addressed my key criticisms. I therefore suggest below how they may address them.

Major

In answer to my first comment, the authors tested the robustness of modules to re-rarefaction, i.e. to small noise. This is an important test to carry out and its outcome strengthens the results of the authors. However, this was not my point. My point was that presence and absence of OTUs on hosts depends on sequencing depth; thus a network built from presence/absence data may differ if it had been constructed for a different sequencing depth. To address this remark, the authors need to discuss this problem in the manuscript and check the robustness of their results to different rarefaction depths, or alternatively build a network from relative abundance data, as suggested below.

The application of relative abundances as a filter criterion does not solve the problem - presence/absences of remaining OTUs are still dependent on sequencing depth.

"we believe these further analyses are outside the scope of the present work, since they imply the construction and analysis of a different network: an OTU-OTU network, which will exclude the sponge hosts from the network analysis"

The host type (HMA/LMA) can be taken into account as a binary variable during the OTU-OTU network construction. In addition to host type, other environmental data such as oceanic region (also as binary variables) or temperature can be integrated in such a network. Microbes associated to different host types could then be identified through network clustering or simply as first-degree neighbours of the host type variable. Network clusters could be tested for significant enrichment of OTUs unique for a particular host species or an oceanic region (at the current sequencing depth). Alternatively, the authors could build networks separately for the two host types in different oceanic regions and compare their properties.

Compared to host-microbe bipartite networks built from presence/absence data, this approach has the advantage of fully exploiting the relative abundances and environmental data and moreover it is less dependent on sequencing depth. So it is fully within the scope of this work.

At the risk of bothering the authors with well-known pitfalls, I would still like to emphasise that for microbial association networks, sequencing depth has to be removed (e.g. by rarefaction or normalisation), rare taxa have to be filtered, association p-values have to be corrected for multiple testing and an interpretation of inferred edges as interactions in the absence of further evidence has to be avoided.

Minor

"This is consistent with recent findings by Björk et al., of prevalent ammensal and commensal interactions between microbes in sponge-associated microbial communities"

Previous comment: "Did these authors validate microbial interactions experimentally? If these are only predicted links, there is no guarantee that they represent ecological interactions, since they could be false positives or indirect."

Authors: "In the present work, we do not consider interactions between microbes which is well beyond the scope of the study. The suggested analysis could be done for specific species in a potential future OTU-OTU network analysis as suggested above."

This point was not about interactions in the OTU-OTU network, which can in any case not be proven through network inference only. The point was whether Björk et al. validated their findings experimentally or whether they are based only on predictions, in which case they have to be presented as such.

Previous comment: "Were some sponge species sampled more than once? If yes, was the microbial community conserved within one species?"

Authors: "Yes, as stated in the methods, we selected sponge species for which at least three samples were available. Microbial communities are generally conserved within individuals of the same species (as shown by Thomas et al.4)."

OK, but does this data set support that observation? Some host species are known to have more than one microbial community, for instance the human gut community was proposed to consist of several "enterotypes" (<https://en.wikipedia.org/wiki/Enterotype>).

Authors: "The numbers of the modules are already shown on the x axis of panel A."

Please inform the readers that the x axis applies to all the panels of this figure in the caption.

Concerning Tax4Fun2: Please include a brief description of its principle in the manuscript.

Authors: "Some metabolic pathways retrieved from the functional analysis are outside what we would expect for sponges. However, so as to not arbitrarily remove pathways we considered all results within the analysis."

Of course, but these results should at least be discussed. Else, these bizarre pathways call the entire pathway analysis into question.

L. 76-77: "In general..."

This is not a full sentence.

Response to Reviewer #2 (Remarks to the Author):

The authors have performed valuable additional analyses, but they have not addressed my key criticisms. I therefore suggest below how they may address them.

Major

In answer to my first comment, the authors tested the robustness of modules to re-rarefaction, i.e. to small noise. This is an important test to carry out and its outcome strengthens the results of the authors. However, this was not my point. My point was that presence and absence of OTUs on hosts depends on sequencing depth; thus a network built from presence/absence data may differ if it had been constructed for a different sequencing depth. To address this remark, the authors need to discuss this problem in the manuscript and check the robustness of their results to different rarefaction depths, or alternatively build a network from relative abundance data, as suggested below.

The additional analyses we performed used rarefaction, not re-rarefaction, of the data to prove the robustness of our results and this is by no means testing for 'small noise', as suggested by the reviewer. The rarefaction analysis we performed is a legitimate way of assessing robustness of data to variation in sampling effort (e.g. sequencing depth). Moreover, it fully addresses the main concern expressed in the comment, which is the fact that, quoting from the comment above, '... presence and absence of OTUs on hosts depends on sequencing depth...'. We completely agree with this statement, which is why we investigated and explained in our previous response as follows:

'We rarefied the data to 10^4 reads per sample (i.e., the size of the smallest sample in the dataset) using the `rarefy` function of the `vegan` package. This procedure was repeated independently 100 times to obtain 100 different rarefied instances (i.e., randomisations) of the data.' (extract from previous response to referee).

To clarify this further, rarefaction of the data in this way ensures that all samples have the same sequencing depth (i.e., 10^4 reads, which in our case corresponds to about the smallest sample size) and hence any issues related to differences in sequencing depth are eliminated. Since the procedure of data rarefaction yields different results every time that it is executed (i.e. because the reads chosen for each sample are randomly selected 100 times). Results obtained using this procedure do exclude many of the OTUs present in the entire dataset. Despite this, we show that the modularity partition we detected using the full dataset is still preserved. This means our conclusions are robust and we have fully addressed the reviewers concern.

To reflect this in the manuscript we have incorporated the following text:

'This modular partition of the network is robust to data rarefaction performed independently 100 times to the number of reads in the smallest sample. Subsequent analyses are, therefore, not affected by biases in sequencing depth (Supplementary Table 6).' (lines 155-158), where Supplementary Table 6 is the Table that we presented in the previous rebuttal letter reporting the results of the analyses performed over the rarefied data. This Table has correspondingly been added to the supplementary information file.

This is accompanied by the corresponding description of the data rarefaction procedure in the methods:

'Data rarefaction

To assess the robustness of the modularity partition obtained using the procedure described above to a different sequencing depth, we performed pairwise permutational analyses of variance (PERMANOVA) on 100 rarefied datasets of the original OTU table using the same modules detected in our network as an independent variable. Thus, testing for 'network module' as a source of variation in microbiome composition. In this way, we can assess whether the modular partition obtained for the network (i.e., the main topological feature our analyses focus on) is robust across the rarefied datasets.

We rarefied the data to 10^4 reads per sample (i.e., the size of the smallest sample in the dataset) using the `rarefy` function of the `vegan` package. This procedure was repeated independently 100 times to obtain 100 different rarefied instances (i.e., randomisations) of the data. We then analysed each of these rarefied datasets using pairwise PERMANOVA with the `adonis` function from `vegan`. P-values were corrected for false discovery rates due to multiple comparisons using the Benjamini-Yekutieli correction. We then obtained the average and

standard deviation of the values obtained from this test across the 100 randomisations of the rarefied data.' (lines 553-569)

The application of relative abundances as a filter criterion does not solve the problem - presence/absences of remaining OTUs are still dependent on sequencing depth.

Again, we agree with this comment, which is why we performed the additional analyses on rarefied data (i.e., removing the issue of sequencing depth). To be as clear as possible, the analyses presented in our previous response are based on rarefied data and we did not rarefy data by relative abundances, as is being suggested in the reviewer's comment. The rarefaction procedure randomly subsampled reads from the full dataset to bring each sample down to the same number of sequences, i.e., we actually discard many sequences chosen at random from each sample to obtain samples with the same sequencing depth. We have therefore completed exactly what the reviewer requested in the first round of comments.

"we believe these further analyses are outside the scope of the present work, since they imply the construction and analysis of a different network: an OTU-OTU network, which will exclude the sponge hosts from the network analysis"

The host type (HMA/LMA) can be taken into account as a binary variable during the OTU-OTU network construction. In addition to host type, other environmental data such as oceanic region (also as binary variables) or temperature can be integrated in such a network. Microbes associated to different host types could then be identified through network clustering or simply as first-degree neighbours of the host type variable. Network clusters could be tested for significant enrichment of OTUs unique for a particular host species or an oceanic region (at the current sequencing depth). Alternatively, the authors could build networks separately for the two host types in different oceanic regions and compare their properties.

Compared to host-microbe bipartite networks built from presence/absence data, this approach has the advantage of fully exploiting the relative abundances and environmental data and moreover it is less dependent on sequencing depth. So it is fully within the scope of this work.

At the risk of bothering the authors with well-known pitfalls, I would still like to emphasise that for microbial association networks, sequencing depth has to be removed (e.g. by rarefaction or normalisation), rare taxa have to be filtered, association p-values have to be corrected for multiple testing and an interpretation of inferred edges as interactions in the absence of further evidence has to be avoided.

We are not trying to reduce the merit of building a microbe-microbe (OTU-OTU) network for the sponge microbiome. This is a fascinating endeavour and we are actually tackling that problem ourselves elsewhere, in fact considering many of the issues/aspects highlighted by the reviewer above. Considering environmental data and host type into OTU-OTU network construction can facilitate the construction of the network and also help to determine the extent to which the environment plays a role in OTU-OTU associations. However, such aims are different to the aims of the current manuscript. Our aim was to investigate the role of the environment on structuring the host-microbiome network, not the microbe-microbe network.

The primary reason we apply a bipartite host-microbe network is that this type of network is the most appropriate for investigating the influence of host phylogeny on the structuring of the microbiome and the sponge-microbiome network, which is a major focus of this study. For this reason, sponges need to be part of the network. In an OTU-OTU network, host sponge identity cannot be unequivocally assigned to a single module. Thus, results such as those presented in Figures 2 and 3 of our paper would not be possible. Similarly, metabolic functions inferred from the microbiome at the sponge level, as we did here, could not be related to specific modules.

For these reasons, the OTU-OTU network remains outside the scope of our present manuscript as it does not address the primary question of the study.

Minor

"This is consistent with recent findings by Björk et al., of prevalent ammensal and commensal interactions between microbes in sponge-associated microbial communities"□

Previous comment: "Did these authors validate microbial interactions experimentally? If these are only predicted links, there is no guarantee that they represent ecological interactions, since they could be false positives or indirect."

Authors: "In the present work, we do not consider interactions between microbes which is well beyond the scope

of the study. The suggested analysis could be done for specific species in a potential future OTU-OTU network analysis as suggested above."

This point was not about interactions in the OTU-OTU network, which can in any case not be proven through network inference only. The point was whether Björk et al. validated their findings experimentally or whether they are based only on predictions, in which case they have to be presented as such.

It appears we misunderstood the reviewer's original comment, believing he/she was querying whether we had performed any experimental validation within the current study. This has been clarified in the new comment by the reviewer and we can now confirm that Björk et al. did not validate their findings experimentally.

Previous comment: "Were some sponge species sampled more than once? If yes, was the microbial community conserved within one species?"

Authors: "Yes, as stated in the methods, we selected sponge species for which at least three samples were available. Microbial communities are generally conserved within individuals of the same species (as shown by Thomas et al.4)."

OK, but does this data set support that observation? Some host species are known to have more than one microbial community, for instance the human gut community was proposed to consist of several "enterotypes" (<https://en.wikipedia.org/wiki/Enterotype>).

We are aware of the different "enterotypes" found in the human gut, however this pattern does not extend to the sponge microbiome. This is not only supported by data shown in Thomas et al. 2017 (Nature Communications), which constitutes a large subset of the data used in this study, but also by the study of Moitinho-Silva et al. 2017 (Frontiers in Microbiology), where it was shown that the sponge microbiome is consistent across individuals of the same host species,

Authors: "The numbers of the modules are already shown on the x axis of panel A."

Please inform the readers that the x axis applies to all the panels of this figure in the caption.

This has been fixed.

Concerning Tax4Fun2: Please include a brief description of its principle in the manuscript.

We have added the following text in the methods section of the manuscript to provide an explanation of how Tax4Fun2 works and was applied in our study:

'Tax4Fun2 extracts functional profiles calculated using complete genomes available through the KEGG database to generate reference data. Functional profiles are obtained from each genome and are then associated with taxonomic keys (i.e., a particular genus) found in the SILVA reference genome database by using the 16S rRNA gene data from each genome. Using sequences aligned against the SILVA database from the original dataset, we used Tax4Fun2 to check whether a reference profile was available for each taxonomic classification. Tax4Fun2 then calculated a predicted metagenome incorporating the abundance of each OTU. OTUs assigned to a taxonomic key having no functional reference profiles would not be included in the prediction (called Fraction of Unexplained; FTU).

For the present study, we generated the reference data as follows: we obtained more than 10,000 complete genomes available in RefSeq (the NCBI reference sequence database) and inferred a functional profile for each based on KEGG KO. 16S rRNA gene sequences from each genome were extracted. Functional profiles of each genome were normalised by the number of 16S rRNA genes within the genome. Extracted 16S rRNA gene sequences were clustered at 100% similarity. One sequence of each cluster served as reference for the cluster. A functional reference profile for each 16S rRNA cluster was calculated from all genomes affiliated to a cluster based on the 16S rRNA gene results. To predict a metagenome, OTU sequences were aligned to the reference 16S rRNA gene sequences with an identity cut-off of 97% (this threshold can be decreased, but will reduce accuracy). OTUs with a lower identity were not considered in the downstream calculation. The abundance of the remaining OTUs in each sample and the reference functional profiles of the reference sequence to which the user sequences were associated during the alignment are then matched and an artificial metagenome is thus calculated. The obtained profile is subsequently normalised: the sum of all functions in a sample is 1.' (lines 659-684)

Authors: "Some metabolic pathways retrieved from the functional analysis are outside what we would expect for sponges. However, so as to not arbitrarily remove pathways we considered all results within the analysis."

Of course, but these results should at least be discussed. Else, these bizarre pathways call the entire pathway analysis into question.

We agree with this point highlighted by the reviewer. For this reason, after showing all the 'bizarre' pathways, for which we currently have no ecological or biological explanation, we focus on pathways involved in the production of bioactive secondary metabolites, which are commonly produced by sponges and their associated microbiomes (see literature reference in the manuscript).

We have a paragraph in the results discussing this:

'12 (or 14%) of the functions that differentiated between modules included pathways for the production of bioactive secondary metabolites; for example, streptomycin biosynthesis, polyketide sugar unit biosynthesis, sesquiterpenoid and triterpenoid biosynthesis, isoflavonoid biosynthesis, antibiotic biosynthesis, prodigiosin biosynthesis and biosynthesis of enediynes antibiotics (Supplementary Figure 5). Sponges and their microbial symbionts are renowned for their bioactive compound production^{38,39} and this function appears to have a significant role in structuring the microbiome.' (lines 293-300)

We also refer to this in the discussion.

L. 76-77: "In general..." This is not a full sentence.

We have changed the sentence to: 'In general, HMA sponges harbour more stable and diverse microbial communities than LMA sponges.' (lines 85-86)

Reviewers' Comments:

Reviewer #2:

Remarks to the Author:

The response of the authors now convinced me that their work is robust to a change in sequencing depth and thus my main criticism is addressed. I admit that last time I overlooked that the re-rarefactions were carried out for 10^4 reads instead of the original 10^5 and excuse for my oversight. In any case, it is a good idea that the authors now describe this analysis in the main manuscript, as other readers may raise a similar concern.

"We are not trying to reduce the merit of building a microbe-microbe (OTU-OTU) network for the sponge microbiome. This is a fascinating endeavour and we are actually tackling that problem ourselves elsewhere, in fact considering many of the issues/aspects highlighted by the reviewer above."

Personally, I am surprised about the refusal to construct OTU-OTU networks in addition to the OTU-host networks, given that such an analysis is likely to offer additional insight into the assembly of sponge microbiota. When targeting a high-impact-factor journal such as Nature Communications, it should be normal to carry out an exhaustive analysis of the data to answer the scientific question of interest, especially for pure analysis papers. However, since I do agree that it is debatable whether abundance-based microbial network inference and analysis is necessary for the study of sponge microbial assembly, I am willing to respect the opinion of the authors on this point.

Instead, I strongly recommend the authors to perform at least a standard multivariate analysis of the microbial profiles in a sample-wise manner, such as a PCoA or equivalent. Such an analysis would allow testing whether individuals of the same host species cluster together in terms of their microbial abundance profiles (not only microbial presence/absence), whether some host species/host types have a more variable microbiota than others and whether clustering of samples is driven by host taxonomy/type, geographic region or other environmental variables (biplot). The authors performed a similar analysis previously (<https://www.ncbi.nlm.nih.gov/pmc/articles/PMC6025780/>), but the current data set is far larger (2000 samples from 150 sponge species instead of 90 samples from 20 sponge species), so it makes sense to repeat it here.

REVIEWERS' COMMENTS

Reviewer #2 (Remarks to the Author):

The response of the authors now convinced me that their work is robust to a change in sequencing depth and thus my main criticism is addressed. I admit that last time I overlooked that the re-rarefactions were carried out for 10^4 reads instead of the original 10^5 and excuse for my oversight. In any case, it is a good idea that the authors now describe this analysis in the main manuscript, as other readers may raise a similar concern.

We are pleased to see that the reviewer is satisfied with this way of performing re-rarefaction.

"We are not trying to reduce the merit of building a microbe-microbe (OTU-OTU) network for the sponge microbiome. This is a fascinating endeavour and we are actually tackling that problem ourselves elsewhere, in fact considering many of the issues/aspects highlighted by the reviewer above."

Personally, I am surprised about the refusal to construct OTU-OTU networks in addition to the OTU-host networks, given that such an analysis is likely to offer additional insight into the assembly of sponge microbiota. When targeting a high-impact-factor journal such as Nature Communications, it should be normal to carry out an exhaustive analysis of the data to answer the scientific question of interest, especially for pure analysis papers. However, since I do agree that it is debatable whether abundance-based microbial network inference and analysis is necessary for the study of sponge microbial assembly, I am willing to respect the opinion of the authors on this point.

Instead, I strongly recommend the authors to perform at least a standard multivariate analysis of the microbial profiles in a sample-wise manner, such as a PCoA or equivalent. Such an analysis would allow testing whether individuals of the same host species cluster together in terms of their microbial abundance profiles (not only microbial presence/absence), whether some host species/host types have a more variable microbiota than others and whether clustering of samples is driven by host taxonomy/type, geographic region or other environmental variables (biplot). The authors performed a similar analysis previously (<https://www.ncbi.nlm.nih.gov/pmc/articles/PMC6025780/>), but the current data set is far larger (2000 samples from 150 sponge species instead of 90 samples from 20 sponge species), so it makes sense to repeat it here.

We agree with the reviewer that such a standard ordination analysis of the sample-level microbial profiles is interesting for exploring patterns of microbiome similarities across sponge species. Along with tests for fitting environmental/explanatory variables to the ordination, this could also reveal the influence of these on microbiome similarity.

Initially, it didn't occur to us that this analysis would add much value to the present manuscript because it has been already done in a previous publication by Thomas et al. (<https://www.nature.com/articles/ncomms11870>), over a subset of this data containing 804 samples from 81 species. These dataset was, in a later iteration of the Sponge Microbiome Project, updated to the one used here.

In any case, we decided to have a look at the new ordination of the samples but, and in order to make it comparable with the previous paper, we performed the ordination using non-metric multi-dimensional scaling (NMDS) of Bray-Curtis distances. This, as anticipated by the reviewer, shows (in the same way it did in our previous publication) that some host species have a more variable microbiota than others, and that the composition of the microbiota is highly determined by sponge species identity. To assess the extent to which this clustering is driven by host identity, environmental factors or sponge traits, we fitted the predictor variables (i.e., host species name, marine ecoregion, sampling depth, temperature, and host type) into the ordination. This analysis revealed that the goodness of fit (GoF, a measure akin to the proportion of the variability explained) of the host species identity ($r^2 = 0.9025$) is substantially higher than that of the other predictor variables: GoF = $2e-4$, 0.033, 0.0706 and 0.3141 for host type, water temperature, depth, and ecoregion respectively.

We have added this analysis to the main text and the NMDS figure to the supplementary material. The following text has been added to the results: 'Evolution of the microbiome within its host is also supported by evidence of high specificity of microbial composition across host species (Supplementary Fig. 8).' (lines 304-306)

Additionally, we added the following description of this analysis to the methods section: 'Non-metric multidimensional scaling (NMDS) to obtain the ordination of community similarity was performed using the metaMDS function of the vegan package with Bray-Curtis as the dissimilarity measure. Fitting of the environmental factors and sponge traits to this ordination was conducted using the envfit function from vegan. R2 (i.e., goodness of fit) was used to assess the ability of the predictor variables to explain the variability of the microbial communities.' (lines 926-932).